# Optogenetic inhibition of actomyosin reveals mechanical bistability of the mesoderm epithelium during *Drosophila* mesoderm invagination

**Hanqing Guo[1], Michael Swan[2], Bing He[1]***

[1]Department of Biological Sciences, Dartmouth College, Hanover, United States; [2]Department of Molecular Biology, Princeton University, Princeton, United States

**Abstract** Apical constriction driven by actin and non-muscle myosin II (actomyosin) provides a well-conserved mechanism to mediate epithelial folding. It remains unclear how contractile forces near the apical surface of a cell sheet drive out-of-the-plane bending of the sheet and whether myosin contractility is required throughout folding. By optogenetic-mediated acute inhibition of actomyosin, we find that during *Drosophila* mesoderm invagination, actomyosin contractility is critical to prevent tissue relaxation during the early, 'priming' stage of folding but is dispensable for the actual folding step after the tissue passes through a stereotyped transitional configuration. This binary response suggests that *Drosophila* mesoderm is mechanically bistable during gastrulation. Computer modeling analysis demonstrates that the binary tissue response to actomyosin inhibition can be recapitulated in the simulated epithelium that undergoes buckling-like deformation jointly mediated by apical constriction in the mesoderm and in-plane compression generated by apico-basal shrinkage of the surrounding ectoderm. Interestingly, comparison between wild-type and *snail* mutants that fail to specify the mesoderm demonstrates that the lateral ectoderm undergoes apicobasal shrinkage during gastrulation independently of mesoderm invagination. We propose that *Drosophila* mesoderm invagination is achieved through an interplay between local apical constriction and mechanical bistability of the epithelium that facilitates epithelial buckling.

*For correspondence:
Bing.He@Dartmouth.edu

Competing interest: The authors declare that no competing interests exist.

## Editor's evaluation

The authors examine the process of mesoderm invagination in the *Drosphila* embryo and found that while myosin contractility is critical to prevent tissue relaxation during the early phase of the process, it is dispensable for the subsequent folding step. Through modeling and experimental analyses, the authors find that folding is likely mediated by a joint action of active cell shape changes in the mesoderm and apico-basal shrinking in the surrounding ectoderm and suggest that the mesoderm behave as a mechanically bistable tissue during gastrulation.

## Introduction

Contractile forces generated by actin and myosin (actomyosin) networks are widely employed in embryogenesis to drive cell motility and cell shape change that pattern epithelial tissues (*Martin and Goldstein, 2014*; *Munjal and Lecuit, 2014*). During epithelial folding mediated by apical constriction, the actomyosin network in epithelial cells constricts cell apices, which leads to bending of the cell sheet (*Sawyer et al., 2010*). It remains unclear how 'in-plane' contractile forces generated at the apical surface drive 'out-of-the-plane' folding of the tissue. The invagination of the presumptive

mesoderm during *Drosophila* gastrulation is a well-characterized epithelial folding process mediated by apical constriction (reviewed in *Gilmour et al., 2017*; *Martin, 2020*; *Gheisari et al., 2020*). During gastrulation, ventrally localized mesoderm precursor cells constrict their cell apices and become internalized into a ventral furrow. The folding process occurs in two steps (*Leptin and Grunewald, 1990*; *Sweeton et al., 1991*). During the first 10–12 min, the ventral cells undergo apical constriction and elongate in the apical-basal direction, while the cell apices remain near the surface of the embryo (the lengthening phase). In the next 6–8 min, the ventral cells rapidly internalize as they shorten back to a wedge-like morphology (the shortening phase) (*Figure 1a*). Ventral furrow formation is controlled by the dorsal-ventral (DV) patterning system. The maternally deposited morphogen Dorsal controls the expression of two transcription factors, Twist and Snail, in the presumptive mesoderm. Twist and Snail in turn cause myosin activation at the apical pole of ventral mesodermal cells through a sequential action of G-protein coupled receptor (GPCR) signaling (*Leptin, 1991*; *Parks and Wieschaus, 1991*; *Costa et al., 1994*; *Kölsch et al., 2007*; *Manning et al., 2013*; *Kerridge et al., 2016*) and RhoGEF2-Rho1-Rho associated kinase (Rok) pathway (*Barrett et al., 1997*; *Häcker and Perrimon, 1998*; *Nikolaidou and Barrett, 2004*; *Dawes-Hoang et al., 2005*; *Martin et al., 2009*; *Mason et al., 2013*). Activated myosin forms a supracellular actomyosin network across the apical surface of the prospective mesoderm and drives the constriction of cell apices (*Martin et al., 2010*; *Martin et al., 2009*; *Martin and Goldstein, 2014*).

The essential role of apical constriction in ventral furrow formation has been well demonstrated. Genetic mutations or pharmacological treatments that inhibit myosin activity disrupt apical constriction and result in failure in ventral furrow formation (reviewed in *Gheisari et al., 2020*; *Martin, 2020*). Biophysical studies show that cell shape change occurred during the lengthening phase is a direct, viscous response of the tissue interior to apical constriction (*Gelbart et al., 2012*; *He et al., 2014*). However, accumulating evidence suggests that apical constriction does not directly drive invagination during the shortening phase. First, it has been observed that the maximal rate of apical constriction (or cell lengthening) and the maximal rate of tissue invagination occur at distinct times (*Polyakov et al., 2014*; *Rauzi et al., 2015*), and timed injection of Rok inhibitor indicates that the late stage of ventral furrow invagination is less sensitive to myosin inactivation (*Krajcovic and Minden, 2012*). Second, it has been previously proposed, and more recently experimentally demonstrated, that myosin accumulated at the lateral membranes of constricting cells (lateral myosin) facilitates furrow invagination by exerting tension along the apical-basal axis of the cell (*Brodland et al., 2010*; *Conte et al., 2012*; *Gracia et al., 2019*; *John and Rauzi, 2021*). Finally, a number of computational models predict that mesoderm invagination requires additional mechanical input from outside of the mesoderm, such as 'pushing' forces from the surrounding ectodermal tissue (*Muñoz et al., 2007*; *Conte et al., 2009*; *Allena et al., 2010*; *Brodland et al., 2010*). These models are in line with the finding that blocking the movement of the lateral ectoderm by laser cauterization inhibits mesoderm invagination (*Rauzi et al., 2015*). A similar disruption of ventral furrow formation can also be achieved by increasing actomyosin contractility in the lateral ectoderm (*Perez-Mockus et al., 2017*). While these pioneer studies highlight the importance of cross-tissue coordination during mesoderm invagination, the actual mechanical mechanism that drives the folding of the mesodermal epithelium and the potential role of the surrounding ectodermal tissue remain to be elucidated.

In this study, we investigate the mechanics of ventral furrow formation by asking whether actomyosin contractility is required throughout the folding process. By developing an optogenetic tool to acutely inhibit actomyosin activity, we find that the dependence of furrow invagination on actomyosin contractility is strongly stage-dependent: Inhibition of actomyosin during apical constriction results in immediate relaxation of the constricted tissue, whereas similar treatment after a stereotyped transitional stage does not impede invagination, suggesting that the mesoderm epithelium has two stable mechanical status during gastrulation. The binary tissue response to actomyosin inhibition can be recapitulated by a 2D vertex model that combines apical constriction in the mesoderm and apicobasal shortening in the neighboring ectoderm, which generates in-plane compression due to volume conservation. Finally, we show evidence the lateral ectoderm undergoes apicobasal shortening around the transitional stage of ventral furrow formation. This ectodermal shortening process, as well as the associated ventrally directed movement of the ectoderm, could occur independently of mesoderm invagination. Taken together, we propose that *Drosophila* mesoderm epithelium is mechanically bistable during gastrulation. We further hypothesize that the mechanical bistability of the mesoderm

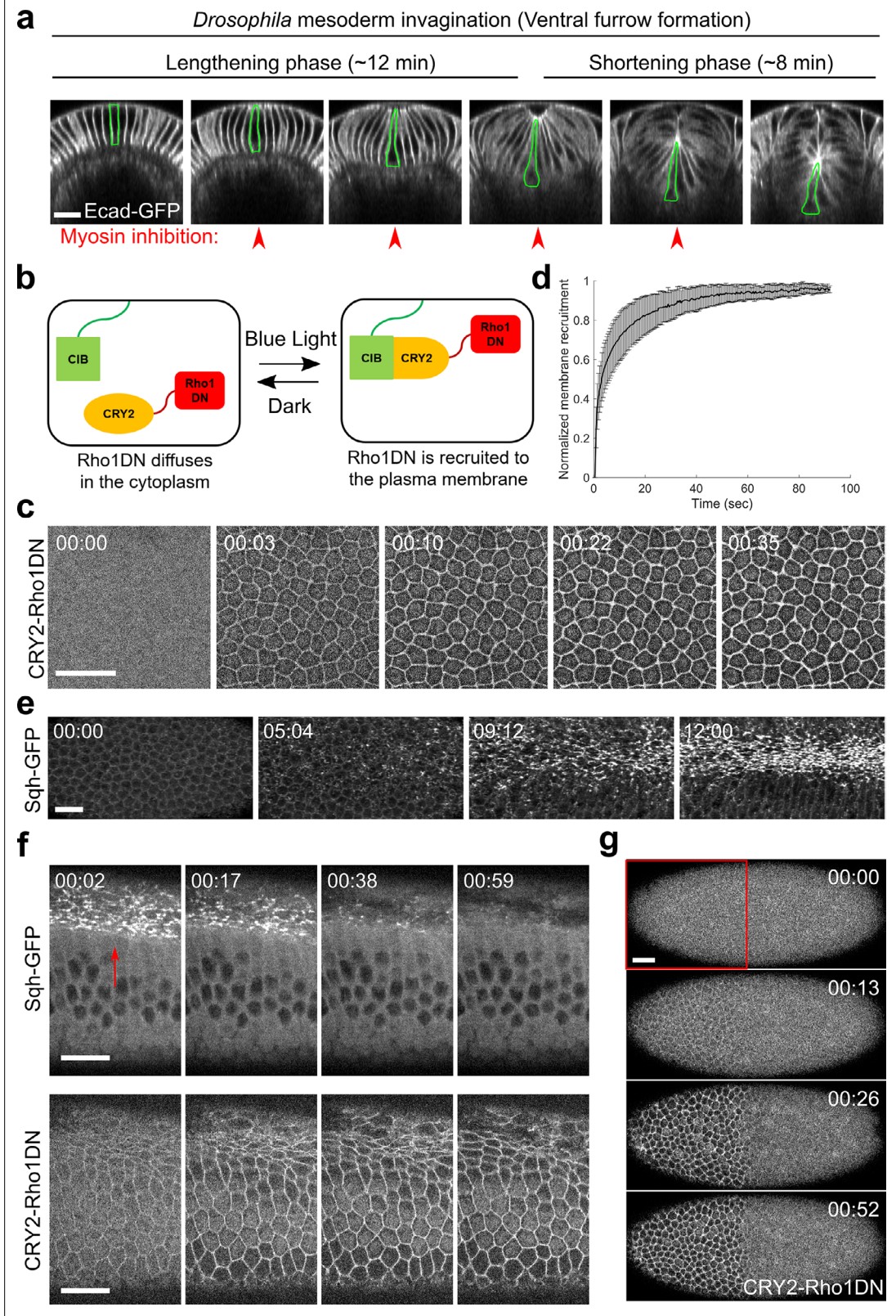

**Figure 1.** Light-dependent membrane recruitment of CRY2-Rho1DN results in rapid myosin inactivation. (**a**) *Drosophila* mesoderm invagination occurs in distinct lengthening and shortening phases. Ventral side is up (same for all cross-section images in this work unless otherwise stated). A single ventral cell undergoing apical constriction is outlined in green. In this study, we sought to test whether myosin contractility is required throughout the folding process by acute, stage-specific inhibition of Rho1 (arrowheads). (**b**) Cartoon depicting the principle of the optogenetic tool used in this study. Upon

*Figure 1 continued on next page*

*Figure 1 continued*

blue light stimulation, CRY2-Rho1DN is translocated from the cytosol to the plasma membrane through the interaction between CRY2 and membrane anchored CIBN. (**c**) Confocal images showing the rapid membrane recruitment of CRY2-Rho1DN upon blue light illumination. (**d**) Relative abundance of membrane recruited CRY2-Rho1DN over time after blue light stimulation. Error bar: s.d., N=6 embryos. (**e**) A wild-type embryo expressing Sqh-GFP showing apical myosin accumulation during ventral furrow formation. N=4 embryos. (**f**) Activation of Opto-Rho1DN results in rapid dissociation of myosin from the ventral cell cortex (arrow) in a gastrulating embryo. N=8 embryos. (**g**) Confocal images showing the confined membrane recruitment of CRY2-Rho1DN within a region of interest (ROI, red box) that has been scanned by a focused beam of blue laser. N=6 embryos. All scale bars=20 µm.

The online version of this article includes the following video and figure supplement(s) for figure 1:

**Figure supplement 1.** Activation of Opto-Rho1DN inhibits apical constriction during gastrulation.

**Figure 1—video 1.** Rapid membrane recruitment of CRY2-Rho1DN upon blue light irradiation.

https://elifesciences.org/articles/69082/figures#fig1video1

**Figure 1—video 2.** Loss of apical myosin during apical constriction upon activation of Opto-Rho1DN.

https://elifesciences.org/articles/69082/figures#fig1video2

is attributed to ectodermal compression and functions together with active cell shape change in the mesoderm to facilitate mesoderm invagination.

## Results

### Plasma membrane recruitment of Rho1DN results in rapid loss of apical myosin and F-actin network

In order to acutely inhibit myosin activity in live embryos (*Figure 1a*), we generated an optogenetic tool, 'Opto-Rho1DN,' to inhibit Rho1 through light-dependent plasma membrane recruitment of a dominant negative form of Rho1 (Rho1DN). Like other Rho GTPases, Rho1 cycles between an active, GTP-bound state and an inactive, GDP-bound state (*Etienne-Manneville and Hall, 2002*). Rho1DN bears a T19N mutation, which abolishes the ability of the mutant protein to exchange GDP for GTP and thereby locks the protein in the inactive state (*Barrett et al., 1997*). A second mutation in Rho1DN, C189Y, abolishes its naive membrane targeting signal (*Sebti and Der, 2003*; *Roberts et al., 2008*). When Rho1DN is recruited to the plasma membrane, it binds to and sequesters Rho1 GEFs, thereby preventing activation of endogenous Rho1 as well as Rho1-Rok-mediated activation of myosin (*Barrett et al., 1997*; *Feig and Cooper, 1988*).

Opto-Rho1DN is based on the CRY2-CIB light-sensitive dimerization system (*Liu et al., 2008*; *Kennedy et al., 2010*; *Guglielmi et al., 2015*; *Figure 1b*). The tool contains two components: a GFP-tagged N-terminal of CIB1 protein (CIBN) anchored at the cytoplasmic leaflet of the plasma membrane (CIBN-pmGFP) (*Guglielmi et al., 2015*), and a fusion protein with Rho1DN, a mCherry tag, and a blue light-reactive protein Cryptochrome 2 (CRY2-Rho1DN). CRY2-Rho1DN remained in the cytoplasm when the sample was kept in the dark or imaged with a yellow-green (561 nm) laser (*Figure 1c*, T=0). Upon blue (488 nm) laser stimulation, CRY2 undergoes a conformational change and binds to CIBN (*Kennedy et al., 2010*; *Liu et al., 2008*), resulting in rapid recruitment of CRY2-Rho1DN to the plasma membrane (*Figure 1c and d*; *Figure 1—video 1*). The average time for half-maximal membrane recruitment of CRY2-Rho1DN is 4 s (3.9±2.8 s, mean±s.d., n=5 embryos). Embryos expressing CIBN-pmGFP and CRY2-Rho1DN developed normally in a dark environment and could hatch (100%, n=63 embryos). In contrast, most embryos (>85%, n=70 embryos) kept in ambient room light failed to hatch, reflecting the essential role of Rho1 in *Drosophila* embryogenesis (*Barrett et al., 1997*; *Magie et al., 1999*; *Johndrow et al., 2004*).

The effectiveness of Opto-Rho1DN on myosin inactivation was validated by stimulating embryos during apical constriction, which resulted in diminishing of apical myosin signal within 60 s (*Figure 1e and f*; *Figure 1—video 2*). Rapid disappearance of cortical myosin upon Rho1 inhibition is consistent with the notion that Rho1 and myosin cycle quickly through active and inactive statues due to the activity of GTPase activating proteins (GAPs) for Rho1 and myosin light chain phosphatases, respectively (*Munjal et al., 2015*; *Coravos and Martin, 2016*). Our data also provide direct evidence that sustained myosin activity during ventral furrow formation requires persistent activation of Rho1.

In addition to myosin, we found that acute Rho1 inhibition also affects cortical F-actin networks during apical constriction, presumably through inactivation of the Rho1 effector formin Diaphanous

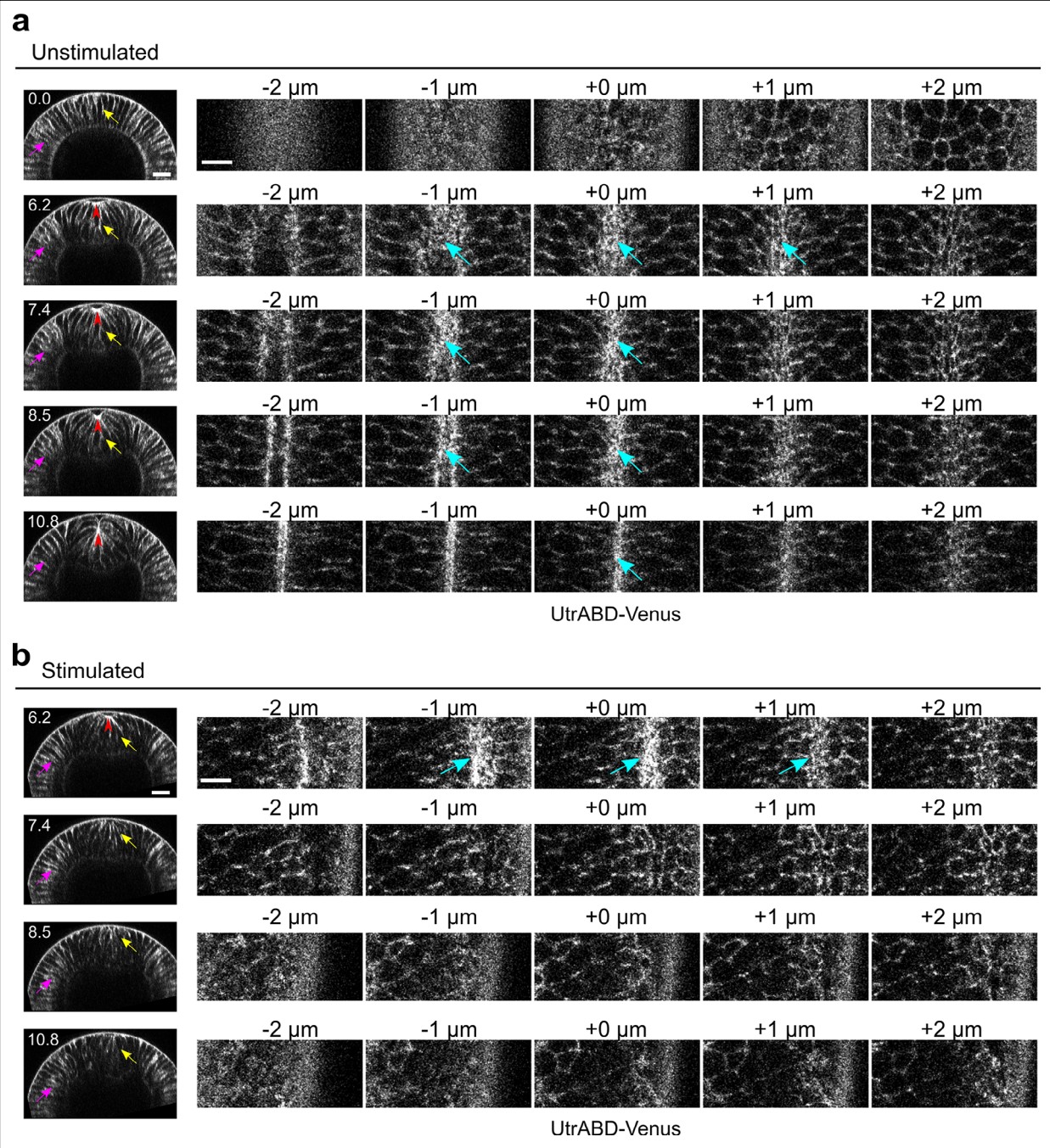

**Figure 2.** Rapid disassembly of apical F-actin in the constricting cells after Opto-Rho1DN stimulation. (**a**) Cross-sections (left) and en face views (right) of a representative unstimulated embryo showing the localization of the F-actin marker UtrophinABD-Venus (UtrABD-Venus) during ventral furrow formation. At the onset of apical constriction, F-actin is enriched along the lateral membrane in both mesodermal and ectodermal cells (yellow and magenta arrows, respectively). During apical constriction, F-actin also accumulates at the apical domain of the constricting cells (cyan arrows). T=0 (min) is the onset of ventral furrow formation. +0 μm indicates the apex of the ventral most cells (red arrowheads) where the strongest accumulation of apical F-actin is observed. (**b**) Upon Opto-Rho1DN stimulation (immediately before 6.2 min), apical F-actin disappears within 1.2 min (cyan arrows). Lateral F-actin in the constricting cells was not immediately affected and only started to diminish 4 min after stimulation (yellow arrows). Lateral F-actin in the ectodermal cells was not significantly affected (magenta arrows). Note that the initial apical indentation (red arrowheads) quickly disappeared after stimulation upon relaxation of the cell apex. N=3 for unstimulated embryos and N=10 for stimulated embryos. Scale bars: cross-sections, 20 μm; en face views, 10 μm.

(*Johndrow et al., 2004*; *Homem and Peifer, 2008*; *Mason et al., 2013*; *Coravos and Martin, 2016*). In unstimulated embryo, F-actin is enriched at the apical domain of the constricting cells (*Figure 2a*, red arrowheads and cyan arrows) and along the lateral membrane in both constricting and non-constricting cells (*Figure 2a*, yellow arrows and magenta arrows). Apical F-actin disappeared within 1.2 min after Opto-Rho1DN stimulation (*Figure 2b*). Lateral F-actin in the constricting cells was not immediately affected and only appeared to diminish 4 min after stimulation (*Figure 2b*). In contrast, lateral F-actin in the ectodermal cells was not significantly affected (*Figure 2b*). Taken together, our results indicate that Opto-Rho1DN is an effective tool for spatially- and temporally-confined inactivation of apical actomyosin contractility through simultaneous myosin inactivation and actin disassembly.

Consistent with the actomyosin phenotype, Opto-Rho1DN stimulation before or during apical constriction resulted in prevention or reversal of apical constriction, respectively (*Figure 1—figure supplement 1*). In addition, Opto-Rho1DN stimulation resulted in an immediate loss of cortical tension at the ventral surface of the embryo during apical constriction (*Figure 3*), in accordance with the previous finding that the increase in tissue tension in the ventral mesodermal region during apical constriction is due to activation of apical myosin contractility and integration of contractile forces across the constriction domain (*Martin et al., 2010*). Finally, by stimulating the embryo within a specific region of interest (ROI), we showed that membrane recruitment of CRY2-Rho1DN and inhibition of apical constriction are restricted to cells within the stimulated region (*Figure 1g*, *Figure 1—figure supplement 1c, d*).

## Acute inhibition of myosin contractility reveals mechanical bistability of the mesoderm during gastrulation

Using Opto-Rho1DN, we determined the role of actomyosin contractility at different stages of ventral furrow formation. We first obtained non-stimulated control movies by imaging embryos bearing Opto-Rho1DN on a multiphoton microscope using a 1040-nm pulsed laser, which excites mCherry but does not stimulate CRY2 (*Guglielmi et al., 2015*). Without stimulation, the embryos underwent normal ventral furrow formation (*Figure 4a*). The transition from the lengthening phase to the shortening phase ($T_{L\text{-}S\ trans}$) occurred around 10 min after the onset of gastrulation (9.3±1.6 min, mean±s.d., n=10 embryos), which was similar to the wild-type embryos (8.9±0.8 min, mean±s.d., n=6 embryos) (*Figure 4—figure supplement 1a-c*). This transition was characterized by a rapid inward movement of the apically constricted cells and could be readily appreciated by the time evolution of distance between the cell apices and the vitelline membrane of the eggshell (invagination depth 'D,' *Figure 4—figure supplement 1d*).

Next, we examined the effect of Opto-Rho1DN stimulation on furrow invagination (*Figure 4b*; *Figure 4—figure supplement 1e*; see Materials and methods). Measuring the rate of furrow invagination (dD/dt) immediately after stimulation revealed three types of tissue responses to stimulation (*Figure 4a and b*; *Figure 4—figure supplement 1f, g*; *Figure 4—video 1*). For most embryos that were stimulated before T=7 min ('Early Group' embryos), the constricting cells relaxed immediately. The initial apical indentation disappeared (dD/dt<0). The flanking, non-constricting cells, which were previously apically stretched by neighboring constricting cells, shrunk back, indicating that the apical cell cortex was elastic (*Figure 4—figure supplement 2a, b*). In contrast, for most embryos that were stimulated after T=7 min ('Late Group' embryos), furrow invagination proceeded at a normal rate and dD/dt remained positive. Finally, some embryos stimulated around T=7 min (6.5±1.4 min, mean±s.d., n=4 embryos; 'Mid Group' embryos) displayed an intermediate phenotype, where ventral furrow invagination paused for around 5 min (4.7±0.5 min; dD/dt≈0) before invagination resumed. We confirmed that there was no residual apical myosin in Late Group embryos upon stimulation (*Figure 4—figure supplement 3a*). Furthermore, lateral myosin, which has been shown to facilitate furrow invagination (*Gracia et al., 2019*; *John and Rauzi, 2021*), also quickly disappeared after stimulation (*Figure 4—figure supplement 3b, c*). Overall, furrow invagination in Late Group embryos is not due to myosin activities at the apical or lateral cortices of the mesodermal cells. Taken together, these results indicate that actomyosin contractility is required in the early but not the late phase of furrow formation. The narrow time window when Mid Group embryos were stimulated defined the 'transition phase' for sensitivity to myosin inhibition ($T_{trans}$), which is immediately before $T_{L\text{-}S\ trans}$ (*Figure 4c*).

While all Early Group embryos showed tissue relaxation after stimulation, five out of eight of them were able to invaginate after a prolonged delay (*Figure 4b*; *Figure 4—figure supplement 2c-f*).

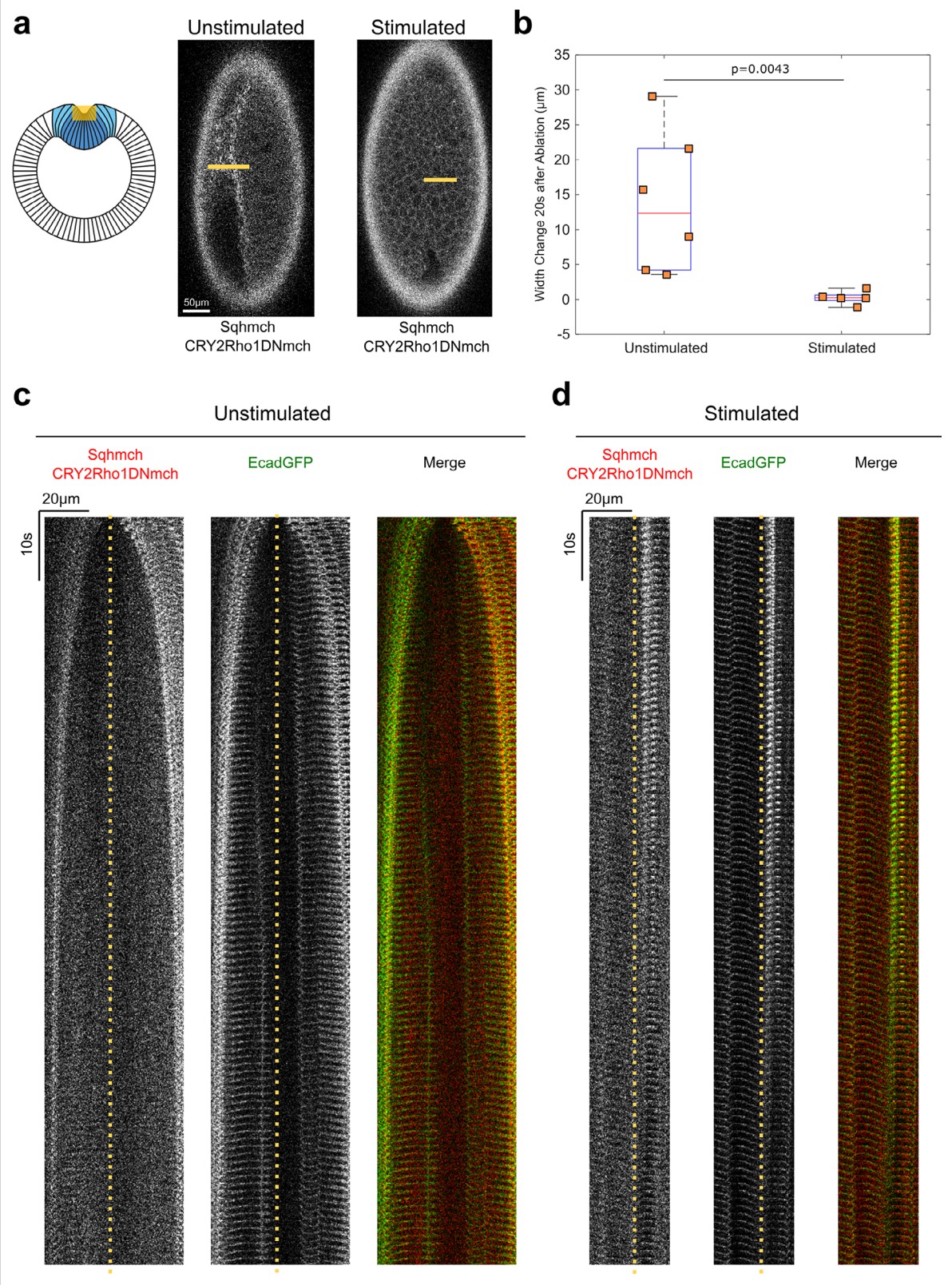

**Figure 3.** Opto-Rho1DN stimulation during apical constriction results in an immediate loss of cortical tension at ventral surface of the embryo. (**a**) Cartoon depicting the experimental setup for laser ablation to detect cortical tension. Yellow shaded regions indicate the ablated regions. For stimulated embryos, light-activation of Opto-Rho1DN was performed 3 min before the laser ablation. Due to apical relaxation after stimulation, multiple z-planes were ablated (yellow shaded region) in order to ensure the ablation of the very apical surface of the ventral cells. (**b**) Width changes of the

*Figure 3 continued on next page*

Figure 3 continued

ablated region along the A-P axis during the first 20 s after laser ablation. A clear tissue recoil was observed after laser cutting in the unstimulated control embryos. In contrast, little to no tissue recoil was observed in the stimulated embryos, indicating lack of apical tension after Rho1 inhibition. p value was calculated using two-sided Wilcoxon rank-sum test. (**c–d**) Kymographs showing the comparison between unstimulated and stimulated embryos. No obvious tissue recoil was observed in the stimulated embryos (N=6 for unstimulated embryos and N=5 for stimulated embryos). Dotted line indicates ablation site.

The invagination was associated with modest myosin reaccumulation at the cell apices that occurred approximately 10 min after the initial stimulation (***Figure 4—figure supplement 2e-f***). It is unclear why apical myosin sometimes reoccurred in the presence of persistent membrane localization of CRY2-Rho1DN. Nevertheless, the completion of furrow invagination when apical constriction is partially impaired is consistent with previous genetic studies and suggests that invagination is facilitated by additional mechanical input (***Costa et al., 1994***; ***Parks and Wieschaus, 1991***).

Because the response of ventral furrow formation to myosin inhibition appears to be binary, we sought to identify morphological features of the progressing furrow that associate with $T_{trans}$. Hence, we examined the relationship between D immediately after stimulation ($D_s$) and the extent by which furrow invagination is affected (assessed by the delay time in furrow invagination, $T_{delay}$; see Materials and methods). We found that for stimulated embryos, $T_{delay}$ was ultrasensitive to $D_s$, with a steep transition at around 6 μm (***Figure 4d***). When $D_s$ was above the threshold, $T_{delay}$ was close to 0. When $D_s$ was below the threshold, $T_{delay}$ increased to 10 min (9.4±1.4 min, mean±s.d., n=5 Early Group embryos) or longer (fail to invaginate, n=3 Early Group embryos). Therefore, the response to acute myosin inhibition can be well predicted by the tissue morphology associated with the stage of furrow progression.

The binary response of the embryo to acute myosin inhibition led us to propose that the mesoderm during gastrulation is mechanically bistable. In this model, the mesoderm epithelium has two stable equilibrium states: the initial, pre-constricted conformation and the final, fully invaginated conformation (***Figure 4e***). Energy input from actomyosin-mediated apical constriction drives the transition from the initial state to an intermediate state ($T_{trans}$), which has the propensity to transition into the final state without further requirement of actomyosin contractility. If myosin is inhibited before $T_{trans}$, the furrow would relax back to the initial state. In contrast, if myosin is inhibited after $T_{trans}$, the furrow would continue to invaginate without pause (***Figure 4e***).

## Analysis of three-dimensional cell shape change after Opto-Rho1DN stimulation

To gain further insights into how cells respond to rapid myosin loss after Rho1 inhibition, we performed three-dimensional cell reconstruction using ilastik (***Berg et al., 2019***). For each embryo, we examined a single row of cells (cells 1–11) along the medial-lateral axis that include the ventral mesodermal cells (cells ~1–6, 'constricting cells'), the lateral mesodermal cells (cells ~7–9, 'non-constricting flanking cells'), and presumably some ectodermal cells (cells ~10 and 11) (***Figure 5a and f***). Our analysis covered time points with most notable cell shape changes after stimulation (0, 2.2, and 4.4 min for Early Group embryos, and 0, 1.1, and 2.2 min for Late Group embryos). Shorter time windows were used for Late group embryos because of the rapid furrow invagination despite the inhibition of actomyosin contractility.

As expected, in Early Group embryos, there was a prompt apical area relaxation in the constricting cells and an accompanying apical area reduction in the flanking cells following stimulation (see ***Figure 5b*** for a cartoon depiction). A number of features were observed in Early Group embryos. First, there was no substantial cell volume change (±5%, ***Figure 5—figure supplement 1***), similar to the observation for normal ventral furrow formation (***Gelbart et al., 2012***; ***Polyakov et al., 2014***). Second, the flanking cells that were most apically stretched before stimulation underwent most prominent apical area reduction, which is anticipated (comparing cells 7–9, ***Figure 5c***). Surprisingly, the extent of apical relaxation in the constriction domain was not uniform. In general, cells located at the side of the constriction domain ('side constricting cells,' cells 4–6) relaxed more than those located closer to the ventral midline ('mid constricting cells,' cells 1–3) (***Figure 5c***). The lower degree of apical relaxation in the mid-constriction cells was accompanied with less prominent 'reversal of lengthening' (***Figure 5d***, see ***Figure 5b*** for a cartoon depiction). Finally, the alteration in apical cell area was associated with changes in cell curvature along the apical-basal axis. Before stimulation, the flanking cells

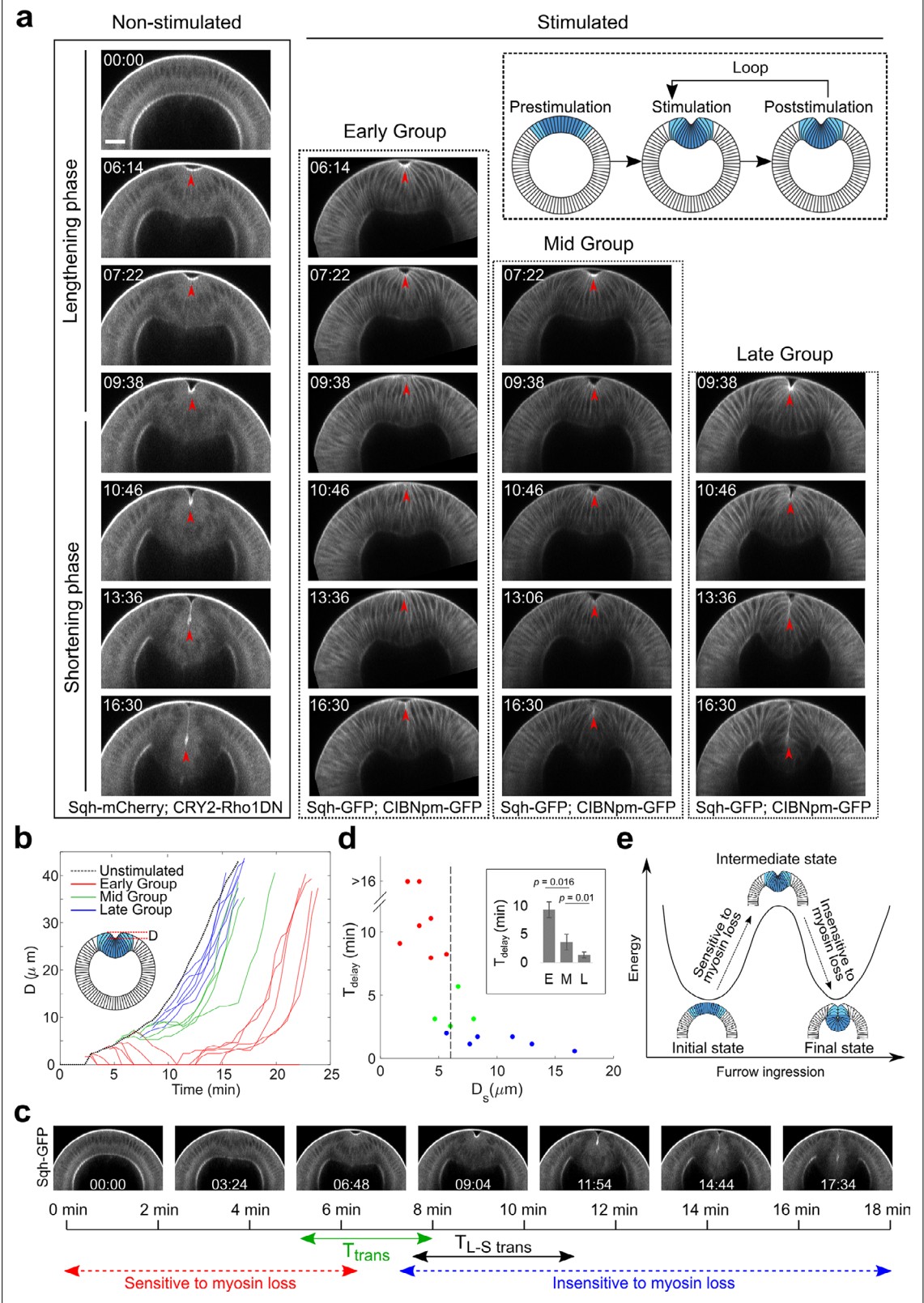

**Figure 4.** Acute inhibition of actomyosin contractility results in stage-dependent response during ventral furrow formation. (**a**) Still images from multiphoton movies showing different tissue responses to acute loss of myosin contractility during ventral furrow formation. Early Group, Mid Group, and Late Group embryos (N=8, 4, and 6 embryos, respectively) are defined based on their immediate response to myosin inhibition. For stimulated embryos, the first frame corresponds to the time point immediately after stimulation. The inset depicts the stimulation and imaging

*Figure 4 continued on next page*

*Figure 4 continued*

protocol. Arrowheads indicate the apex of the ventral most cells. (**b**) Time evolution of the invagination depth 'D' for the stimulated embryos and a representative unstimulated control embryo. For stimulated embryos, all movies were aligned in time to the representative control embryo based on furrow morphology at the time of stimulation. (**c**) Relationship between the transition phase for sensitivity to myosin inhibition ($T_{trans}$) and lengthening-shortening transition ($T_{L\text{-}S\ trans}$). (**d**) Scatter plot showing the relation between invagination depth at the time of stimulation ($D_s$) and the delay time ($T_{delay}$) in furrow invagination compared to the representative control embryo. $T_{delay}$ is highly sensitive to $D_s$, with a switch-like change at $D_s \sim 6\ \mu m$ (dashed line). Inset: Average $T_{delay}$ in Early (E, n=5 embryos that invaginated), Mid (M, n=4), and Late (L, n=6) Group embryos. Statistical comparisons were performed using two-sided Wilcoxon rank-sum test. (**e**) Cartoon depicting mechanical bistability of the mesoderm during gastrulation. Both the initial, pre-constriction state and the final, fully invaginated state are stable. During gastrulation, actomyosin contractility is critical for bringing the system from the initial state to an intermediate, transitional state, whereas the subsequent transition to the final state can occur independent of myosin contractility.

The online version of this article includes the following video, source data, and figure supplement(s) for figure 4:

**Source data 1.** Cell length measurements for determining lengthening-shortening transition time.

**Figure supplement 1.** Classification of the response of embryos to acute myosin inhibition during ventral furrow formation.

**Figure supplement 2.** Tissue response to acute myosin inhibition in Early Group embryos.

**Figure supplement 3.** Dissociation of myosin from both apical and lateral cell cortices in constricting cells upon Opto-Rho1DN stimulation.

**Figure 4—video 1.** Stage-dependent response to acute myosin inhibition during ventral furrow formation.

https://elifesciences.org/articles/69082/figures#fig4video1

and the side constricting cells were all tilted, with their apical side bending toward the ventral midline (e.g., cyan and orange cells in *Figure 5a*). After stimulation, the cells straightened up and partially restored their initial, columnar cell shape. This cell shape change is associated with a reduction in the apical-basal cell length and lateral cell area (*Figure 5b, d and e*; *Figure 5—figure supplement 1*). The ectodermal cells also showed some unbending, but their shape changes were in general moderate (*Figure 5*, cells 10 and 11).

Interestingly, despite the drastic tissue level difference between Early and Late groups of embryos, several trends we observed in Early group was also present in Late group, such as the conservation of cell volume (*Figure 5—figure supplement 1*), the straighten-up of some (but not all) flanking cells (e.g., cell nine in *Figure 5j*), and the uneven pattern of apical relaxation in the constriction domain (*Figure 5h*). Despite these similarities, Late Group embryos displayed several prominent features that were distinct from Early Group embryos (see *Figure 5b and g* for a cartoon depiction). First, the mid constricting cells barely showed any apical relaxation and underwent rapid cell shortening after stimulation (*Figure 5h and i*, cells 1–3), which were not observed in Early Group embryos but rather resembled the normal cell shortening process (*Polyakov et al., 2014*). Second, despite undergoing apical relaxation, the side constricting cells bent further toward the ventral midline instead of straightening up as their counterparts in Early Group embryos (*Figure 5j*, cells 4–6). The behavior of the middle and side constricting cells in Late Group embryos elucidates the cellular basis for the continued deepening of the furrow after Opto-Rho1DN stimulation and explains the tissue level difference between Early and Late Group embryos. Importantly, the observed cell shape change and tissue invagination in Late Group embryos occurred in the absence of actomyosin contractility, suggesting the presence of additional mechanical contributions.

## A vertex model combining apical constriction in the mesoderm and apicobasal shortening in the ectoderm recapitulates the binary response to actomyosin inhibition

What mechanism may account for the apparent mechanical bistability during mesoderm folding? One possible mechanism is germband extension, a body axis elongation process in *Drosophila* that occurs soon after the onset of gastrulation (*Kong et al., 2017*). During germband extension, the ventral and lateral ectoderm undergo convergent extension and converge toward the ventral midline as they extend along the anterior-posterior (AP) axis. If germband extension-mediated ventral movement starts around $T_{trans}$, it may account for the binary tissue response to actomyosin inhibition. However, previous literature indicates that germband extension does not start until ventral furrow formation is nearly completed (*Lye et al., 2015*). Furthermore, germband extension depends on Rho1-mediated activation of actomyosin contractility in the ectodermal cells (*Kong et al., 2017*). Since we stimulated Opto-Rho1DN in both the mesoderm and the ectoderm in our optogenetic experiment, we anticipate

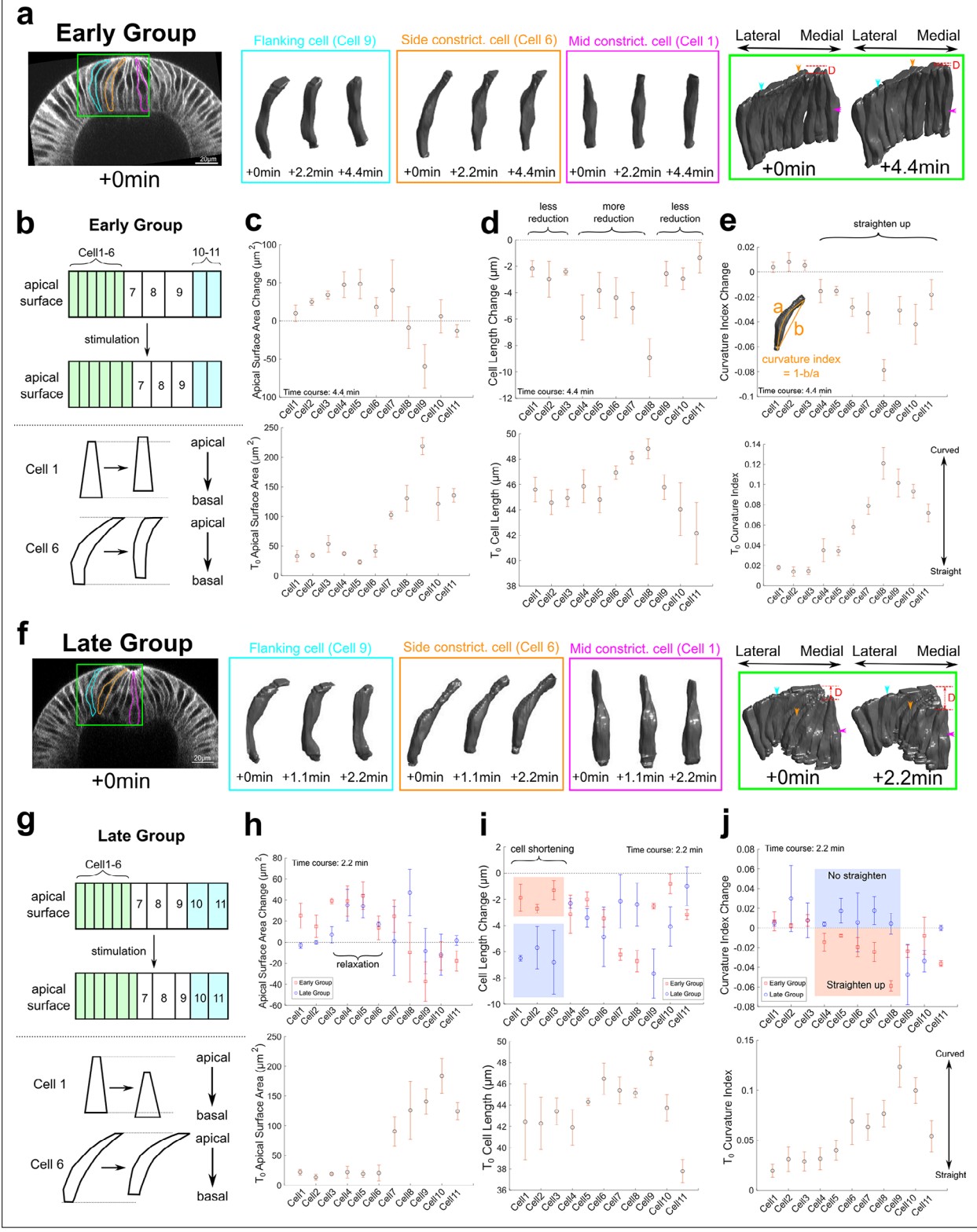

**Figure 5.** Three-dimensional segmentation reveals details of cell response after Opto-Rho1DN stimulation. (**a–e**) Early Group embryos and (**f–j**) Late Group embryos. (**a, f**) A row of cells along the medial-lateral axis on one side of the ventral midline was segmented in 3D (green box) over time after stimulation. Cells 1–6: constricting cells; Cells 7–9: flanking cells; Cells 10 and 11: ectodermal cells. Early Group: T=0, 2.2, and 4.4 min. Late Group: T=0, 1.1, and 2.2 min. 3D rendering of three representative cells, cell 1 (mid constricting cell), cell 6 (side constricting cell), and cell 9 (flanking cell), are shown in magenta, orange, and cyan boxes, respectively. These cells are also marked on the cross-section views (outlines) and the 3D tissue renderings

*Figure 5 continued on next page*

*Figure 5 continued*

(arrowheads) with the same color code. D: invagination depth. (**b, g**) Cartoon depicting apical surface area change (top) and characteristic cell shape changes (bottom) observed in Early Group (**b**) and Late Group (**g**) embryos. (**c–e, h–j**) Upper panels: quantifications showing the apical surface area change (**c, h**), cell length change (**d, i**), and cell curvature change (**e, j**) between 0 and 4.4 min (**c–e**) or between 0 and 2.2 min (**h–j**). Measurements from both Early and Late Groups are shown in (**h-j**) for comparison. Bottom panels show the measurements for each quantity immediately after stimulation (T=0). Error bars: ±s.e. N=3 embryos for each group.

The online version of this article includes the following figure supplement(s) for figure 5:

**Figure supplement 1.** Volume conservation and lateral surface change in Early (**a**) and Late (**b**) Group embryos after Opto-Rho1DN stimulation.

that germband extension would be inhibited. Consistent with this notion, we did not observe AP tissue movement in Opto-Rho1DN treated embryos (*Figure 6—figure supplement 2*), suggesting that other mechanisms are responsible for the invagination of ventral furrow in Late Group embryos.

Previous biophysical studies demonstrate that the embryonic epithelium in early *Drosophila* embryos can display an elastic response with a decay timescale of at least 4 min (*Doubrovinski et al., 2017*). The apparent bistable characteristic of the mesoderm is reminiscent of a compressed elastic beam undergoing snap-through buckling facilitated by a transverse 'indentation force' (*Figure 6b*; *Qiu et al., 2004*). Based on this analogy, we hypothesized that mesoderm invagination is mediated through tissue buckling enabled by in-plane compressive stresses. We tested this hypothesis by implementing compressive stresses from the ectoderm in a previously published two-dimensional vertex model for ventral furrow formation (*Polyakov et al., 2014*; *Figure 6a*; see Materials and methods). In this model, the sole energy input is apical constriction of the ventral cells, whereas the apical, lateral, and basal cortex of cells were represented by elastic springs that resist deformation (*Polyakov et al., 2014*; see Materials and methods). Note that the elasticity assumption in this model is a simplification of the actual viscoelastic properties of the embryonic tissue. However, this model successfully predicted the intermediate and final furrow morphologies with a minimal set of active and passive forces without prescribing individual cell shape changes (*Polyakov et al., 2014*; see Materials and methods). It is therefore advantageous to use this model to explore the main novel aspect of the folding mechanics underlying ventral furrow formation and to test the central concept of our hypothesis based on relatively small number of assumptions.

Using this model, we simulated the effect of acute myosin loss by removing the apical contractile forces at different intermediate states of the model (Materials and methods). Remarkably, in the presence of ectodermal compression, the model recapitulated the binary response to loss of apical constriction as observed experimentally (*Figure 6c*). In particular, the transitional state of the tissue revealed in the simulation is visually similar to that identified in our optogenetic experiments. As a control, in the absence of ectodermal compression, the model always returned to its initial configuration regardless of the stage when apical constriction was eliminated (*Figure 6c*). Therefore, our simulation indicates that buckling of an elastic cell sheet jointly mediated by local apical constriction and global in-plane compression can well reproduce the observed binary tissue response to acute actomyosin inhibition. Our model also predicts that the presence of ectodermal compression could compensate for partial impairment of apical constriction (*Figure 6d*), which explains why some Early Group embryos could still invaginate when myosin activity was greatly downregulated.

Interestingly, our simulation predicts that for embryos stimulated after $T_{trans}$, the ventral furrow at the final invaginated state would be narrower with fewer cells being incorporated into the furrow compared to control embryos. Our observation in Late Group embryos well matched this prediction (*Figure 6c*; *Figure 6—figure supplement 1a, b*). Consistent with our 3D analysis (*Figure 5h*), we found that stimulation in Late Group embryos induced a mild but noticeable alteration in tissue flow that indicates the occurrence of apical relaxation in the constricted cells (*Figure 6—figure supplement 1c, d*). These results suggest that maintaining actomyosin contractility after $T_{trans}$ helps to prevent local relaxation of the apical membranes, thereby allowing more cells to be incorporated into the furrow. This 'late' function of actomyosin contractility is dispensable for furrow invagination per se but is important for the complete internalization of the prospective mesoderm.

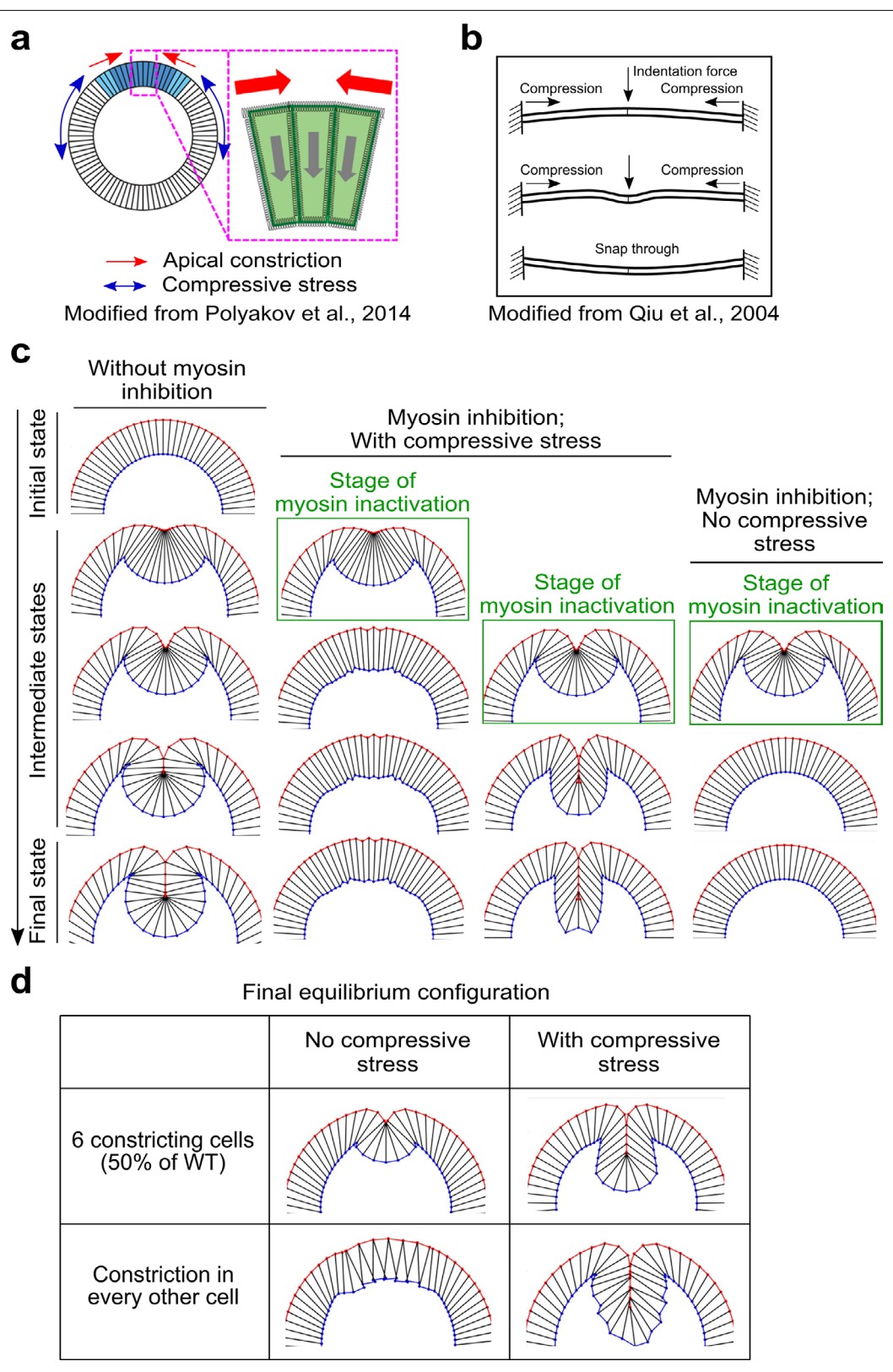

**Figure 6.** Mechanical bistability of the mesoderm during gastrulation can arise from ectoderm-derived compressive stresses. (**a**) 2D vertex model testing the mechanisms of mechanical bistability of the mesoderm during gastrulation. In this model, the apical, lateral, and basal cortices of cells are modeled as elastic springs that resist deformations, the cell interiors are non-compressible, and the only two active energy inputs are apical

*Figure 6 continued*

constriction in the mesoderm (blue cells) and in-plane compression generated by the ectoderm (white cells). (**b**) Buckling of a compressed elastic beam triggered by a transverse indentation force. (**c**) The model recapitulates the bipartite response to acute loss of actomyosin contractility only when the in-plane compression is present. Column 1 shows normal ventral furrow formation. Columns 2–4 show tissue response when apical constriction is inhibited at different stages of ventral furrow formation with or without ectodermal compression. (**d**) Model predictions on the impact of reducing the width of apical constriction domain or the uniformity of apical constriction on final invagination depth. In the presence of compressive stress, final invagination depth becomes less sensitive to perturbations of apical constriction.

The online version of this article includes the following source data and figure supplement(s) for figure 6:

**Source data 1.** Computer code for the energy minimization-based vertex model for ventral furrow formation.

**Figure supplement 1.** Myosin inhibition in Late Group embryos results in fewer cells incorporated into the ventral furrow.

**Figure supplement 2.** Lack of AP movement in Late Group embryos during ventral furrow invagination.

## Presence of myosin-independent cell shortening force in the constriction domain is important for the vertex model to recapitulate the observed binary response

In the vertex model described above, shortening of the constricting cells is mediated by passive elastic restoration forces generated in these cells as their lateral edges are stretched during cell lengthening (*Polyakov et al., 2014*). This passive elastic force is not affected by in silico myosin inactivation. However, our experiments show that Opto-Rho1DN stimulation results in rapid diminishing of lateral myosin in the constricting cells, suggesting that Rho1 inhibition may weaken the shortening forces in these cells (*Gracia et al., 2019*; *John and Rauzi, 2021*). To account for this effect, we implemented active lateral constriction forces in the ventral cells. In such a scenario, the active and passive lateral forces worked in combination to mediate ventral cell shortening, but only the active force was sensitive to myosin inactivation (Materials and methods). The active and passive lateral forces were given by $K_{L\_active}l$ and $K_{L\_passive}(l - l_0)$, respectively, where $l$ and $l_0$ were the current and the resting length of the lateral edge, respectively. As expected, the addition of the active lateral force allows us to reduce the passive lateral force while still generate furrows with normal morphology (*Figure 7a*, comparing column $K_{L\_active}=0$ with column $K_{L\_active}=2$). The implementation of myosin-dependent active lateral forces allowed us to examine how the binary response would be influenced if myosin inactivation weakens the cell shortening forces in the mesoderm.

We performed the tests under the conditions where the model generates normal furrow morphology (green shaded region in *Figure 7a*, $K_{L\_passive} = 20$ while $K_{L\_active}$ ranges from 0 to 2, or $K_{L\_active} = 2$ while $K_{L\_passive}$ ranges from 0.002 to 20). When $K_{L\_passive} = 20$, changing $K_{L\_active}$ has little impact on the response of the model to stimulation (*Figure 7b*). When $K_{L\_active} = 2$, changing $K_{L\_passive}$ did not affect the binary tissue response per se but influenced the final morphology after stimulation (*Figure 7c*). When the passive lateral force was relatively high ($K_{L\_passive} = 20$), the model displayed normal binary response with realistic final furrow morphologies. Lowering the passive lateral force by a factor of ten ($K_{L\_passive} = 2$) did not change the tissue response aside from a mild reduction of the invagination depth 'D' (from 61 µm to 51 µm) in Late Group embryos. Strikingly, further lowering $K_{L\_passive}$ to 0.2 resulted in a very shallow furrow and abnormal constricting cell morphology in Late Group embryos (D=30 µm). Taken together, these results demonstrate that the binary response of the model to myosin inhibition still exists when the cell shortening forces in the mesoderm become sensitive to myosin inhibition. However, a minimal level of myosin-independent lateral elastic force ($K_{L\_passive}>0.2$) is required for the model to generate realistic final furrow morphology in Late Group embryos. Interestingly, we found that in real embryos the lateral actin in the constricting cells persisted for several minutes after Opto-Rho1DN stimulation (*Figure 2b*, yellow arrows), which might provide the lateral elasticity described in our model.

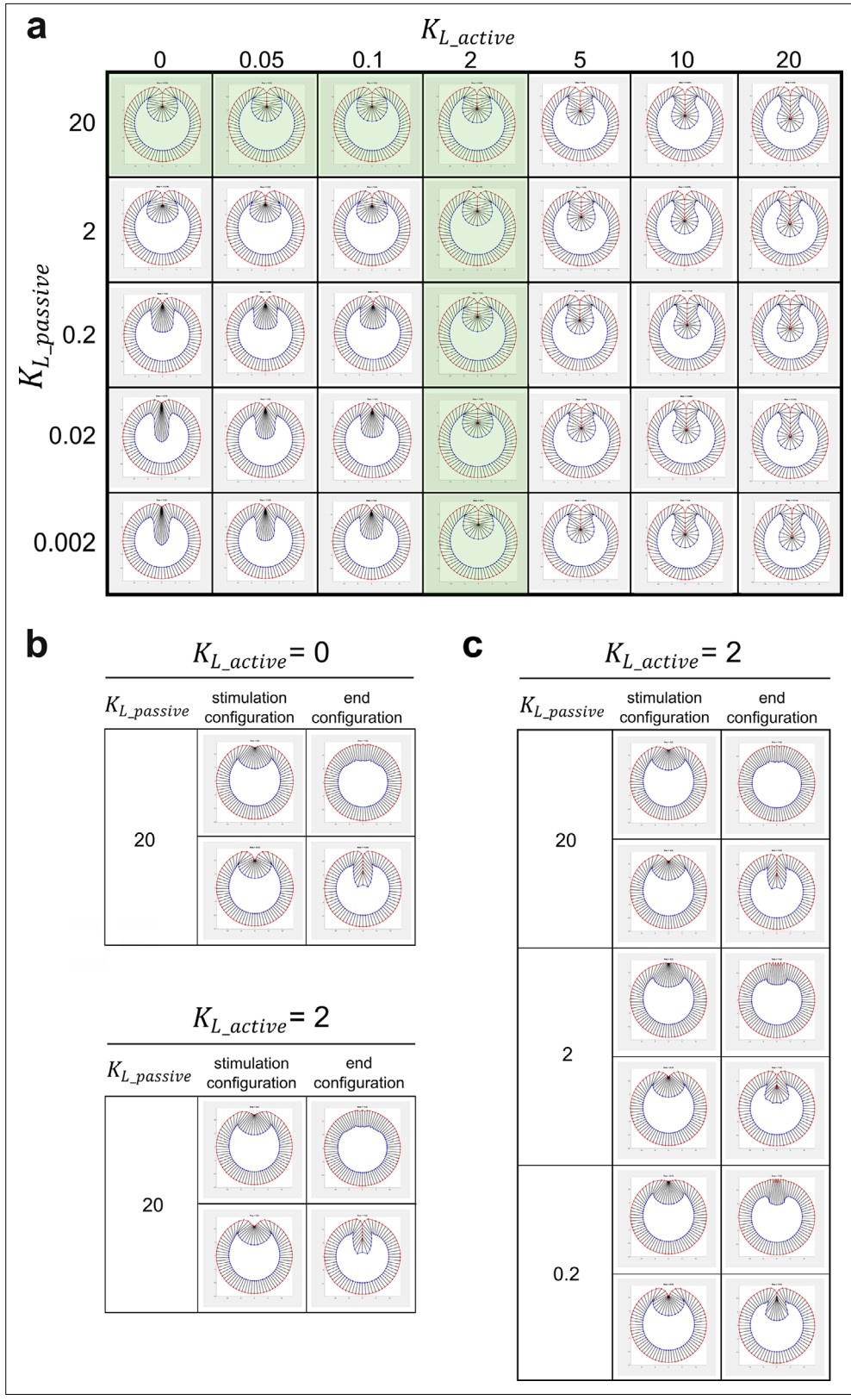

**Figure 7.** Modeling analysis of the impact of myosin-dependent and myosin-independent cell shortening forces in the constriction domain on binary tissue response. (**a**) The myosin-dependent, active lateral shortening forces and myosin-independent, passive lateral shortening forces function additively to mediate cell shortening and furrow invagination in the 2D vertex model. The active and passive lateral forces are given by $K_{L\_active}l$

*Figure 7 continued on next page*

*Figure 7 continued*

and $K_{L\_passive}\,(l-l_0)$, respectively, where $l$ and $l_0$ are the current and the resting length of the lateral edge, respectively. Green shaded region represents the conditions where the model generates normal final furrow morphology. (**b**) When $K_{L\_passive} = 20$, changing $K_{L\_active}$ has little impact on the binary response of the model to stimulation. (**c**) When $K_{L\_active} = 2$, changing $K_{L\_passive}$ does not affect the binary tissue response per se but influences the final morphology after stimulation.

## The lateral ectoderm undergoes apical-basal shortening during gastrulation independently of ventral furrow formation

It remains unclear whether compressive stresses exist in the ectoderm during ventral furrow formation in real embryos. No cell division happens in the embryo during ventral furrow formation (*Foe, 1989*; *Edgar et al., 1994*), thereby excluding a role for cell proliferation in generating compressive stresses as seen in many other tissues (*Nelson, 2016*). Interestingly, it has been reported that ectodermal cells undergo apical-basal shortening during gastrulation (*Brodland et al., 2010*). In theory, this cell shape change could generate compressive stresses in the planar direction if the cell volume remains conserved. To further investigate the extent of ectodermal shortening during gastrulation, we performed 3D tissue volume measurement ($Vol_{ec}$) in the lateral ectodermal region of wild-type embryos (ROI: 60°–120° away from the ventral midline and 75-µm long along the AP axis; *Figure 8a*). We focused our analysis on the lateral ectodermal region since a previous study shows that the dorsal ectoderm does not significantly contribute to ventral furrow formation (*Rauzi et al., 2015*). We reasoned that if the ectodermal cells undergo apical-basal shortening, there should be a net volume outflux from the ROI, causing a reduction in $Vol_{ec}$ (*Figure 8a*). Since cell shortening may not be uniform across the tissue, the change in $Vol_{ec}$ provides a less noisy measure of the change in average tissue thickness. To measure $Vol_{ec}$, we performed 3D tissue segmentation without segmenting individual cells (Materials and methods).

This analysis led to the following observations. First, $Vol_{ec}$ increased during the first 10 min of ventral furrow formation (*Figure 8c*). This observation is consistent with the previous measurement of ectodermal cell length during gastrulation (*Brodland et al., 2010*). Second, in all cases examined, $Vol_{ec}$ started to decrease approximately halfway through ventral furrow formation, but the exact time of this transition varied between embryos (11.1±2.3 min, mean±s.d., n=3 embryos; *Figure 8c and f*, arrowheads). Finally, the percent reduction in $Vol_{ec}$ from the peak volume to that at the end of ventral furrow formation ranged between 4% and 8%. These observations are consistent with an independent 2D measurement of the cross-section area (as a proxy for tissue thickness) of a lateral ectodermal region that was between 60° and 90° from the ventral midline (~6% reduction; *Figure 8—figure supplement 1*).

The observed ectodermal shortening could be driven by an active mechanism that reduces cell length or could be a passive response to the pulling from ventral furrow. To distinguish these possibilities, we repeated the $Vol_{ec}$ measurements in *snail* mutant embryos. As expected, ventral furrow failed to form in *snail* embryos (*Figure 8g*). Strikingly, the reduction of $Vol_{ec}$ occurred at about the same time as in the wild-type. In addition, the rate of $Vol_{ec}$ reduction was similar between the wild-type and *snail* embryos, although the *snail* embryos on average had a smaller $Vol_{ec}$ to start with (*Figure 8d, f and h*). This result indicates that pulling from ventral furrow cannot fully account for the apicobasal shrinkage of lateral ectoderm in the wild-type embryos and suggests the presence of ventral furrow-independent mechanism for ectodermal shortening.

In our vertex model, ectodermal compression was generated by letting all ectodermal cells shorten by 20% (ΔL=20%; Materials and methods). Further analysis demonstrates that a minimum of 5%–10% shortening in all 60 ectodermal cells is critical for generating the binary response, which was greater than the range measured in the actual embryos (4%–8% shortening, measured in ~45% ectodermal cells; *Figure 8*; *Figure 8—figure supplement 2*). A number of factors may account for the difference between the model and the actual embryo. For example, we observed residual cellularization in the ectoderm during the first 10 min of gastrulation, but there was no cell growth in our model (*Figure 8c*). While an apical-basal shrinking of the ectodermal cells can generate in-plane compression, the onset and the effect of this shrinking may be underestimated by volume or length measurement if the cells are growing at the same time. In addition, other morphogenetic processes occurring

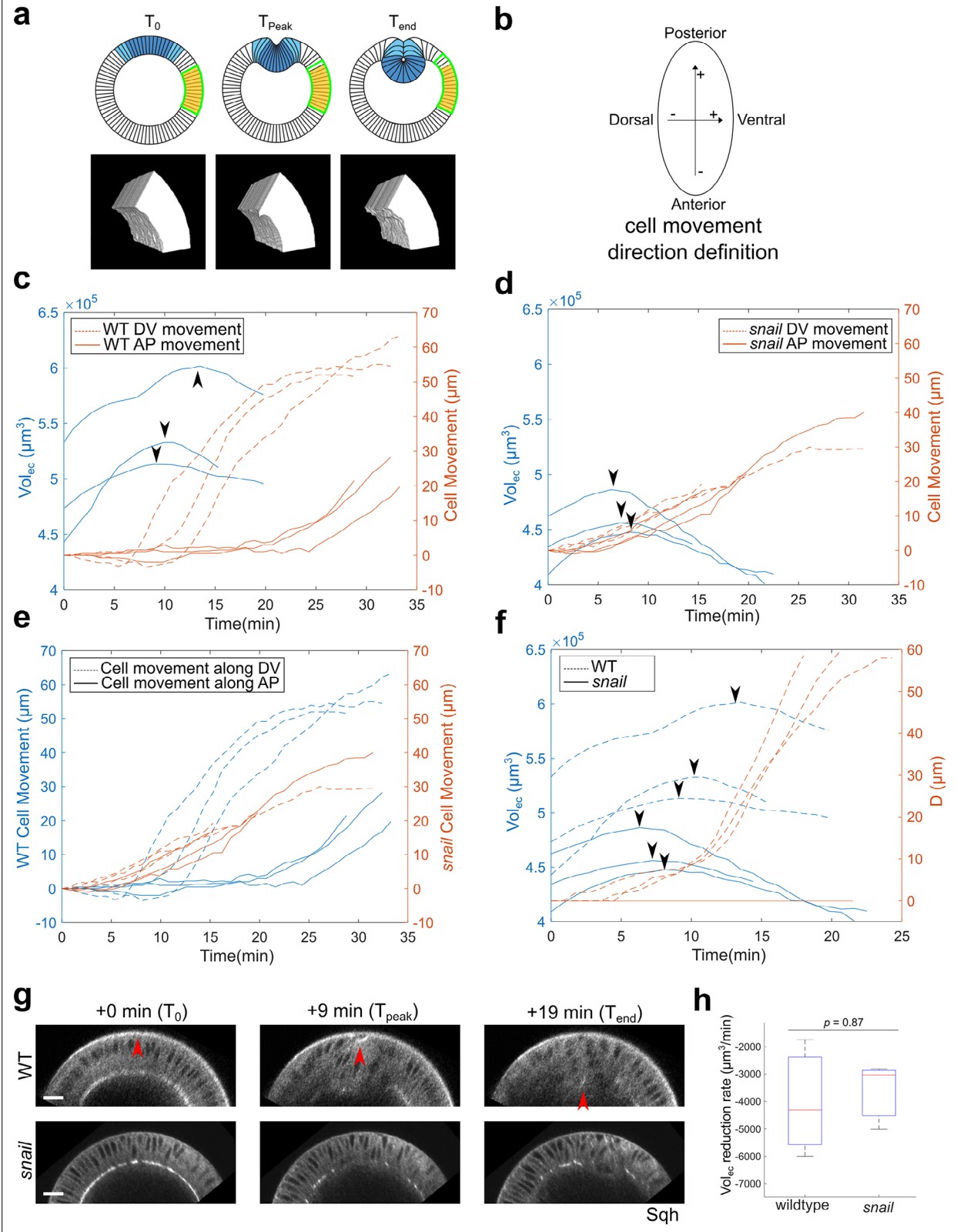

**Figure 8.** The lateral ectoderm undergoes apical-basal shortening during gastrulation independently of ventral furrow formation. (**a**) Top: cartoon showing a cross-section view of an embryo. Blue: mesoderm. Yellow: segmented ectoderm region (ROI). Green: the same group of ectoderm cells. The ROI (in 3D) covers the lateral ectodermal region that is 60°–120° away from the ventral midline and 75-μm long along the AP axis. The change in the volume of ROI ($Vol_{ec}$) is used as a readout for change in average tissue thickness. Bottom: representative segmented 3D views showing the ectoderm

*Figure 8 continued on next page*

*Figure 8 continued*

at the onset of apical constriction ($T_0$), reaches the thickest point ($T_{peak}$) and at the last time frame of ventral furrow formation that is reliably segmented ($T_{end}$). (**b**) Definition for the direction of cell movement used in (**c–f**). (**c, d**) Change in $Vol_{ec}$ over time and cell movement along A-P and D-V axis in wild-type (**c**) and *snail* mutant embryos (**d**). Arrowheads indicate the start of ectoderm shortening. (**e**) Cell movement along A-P and D-V axis in wild-type and *snail* mutants are replotted together for better comparison. (**f**) Change in $Vol_{ec}$ and invagination depth D over time in wild-type and *snail* mutant embryos. Arrowheads indicate the start of ectoderm shortening. (**g**) Representative cross-section views showing the wild-type and *snail* mutant embryos at $T_0$, $T_{peak}$, and $T_{end}$, respectively. Arrowheads indicate the apex of the mid constricting cells. N=3 embryos for each genotype. Scale bars: 20 μm. (**h**) Comparison of the ectoderm volume reduction rate between the wild-type and *snail* mutant embryos. The descending part of the volume curve was fitted into a straight line to calculate the rate of volume reduction.

The online version of this article includes the following figure supplement(s) for figure 8:

**Figure supplement 1.** Lateral ectoderm undergoes apical-basal shortening during ventral furrow formation.

**Figure supplement 2.** The impact of altering the extent of ectodermal shortening on the response of the model to acute myosin inhibition.

**Figure supplement 3.** Measurement of ectoderm cross-section area in Late Group embryos.

during gastrulation might also contribute to the generation of tissue compression. For example, *Rauzi et al., 2015* found that both the onset of rapid ventral furrow invagination and the onset of ventrally directed movement of the lateral ectoderm are delayed in the mutant that disrupts cephalic furrow formation and posterior midgut invagination (*Rauzi et al., 2015*). Finally, the difference between an elastic model and the actual viscoelastic embryonic tissue may also contribute to the discrepancy between the measurements and model predictions. While our model recapitulated the major cell morphological features during ventral furrow formation and correctly predicted the binary response to acute actomyosin inhibition, future research is needed to elucidate the mechanism and function of ectodermal shortening in gastrulation. An interesting prediction from the Late Group embryo observation based on our proposed model is that ectodermal shortening does not depend on Rho1 activity, since ventral furrow invagination continued in Late Group embryos in which both the mesoderm and the ectoderm were stimulated (Materials and methods). Consistent with this prediction, we found that Opto-Rho1DN stimulation only had a mild and transient effect on ectoderm thickness change, suggesting that Rho1 inhibition did not directly impact ectodermal shortening (*Figure 8—figure supplement 3*).

## Ventrally directed movement of the lateral ectoderm still occurs when apical constriction in the mesoderm is inhibited

In theory, compression generated by ectodermal shortening should result in displacement of the ectodermal cells toward the ventral region regardless of whether mesoderm invagination occurs. Previous studies have shown that ventral movement of the lateral ectoderm still occurs in mesoderm specification mutants that do not form ventral furrow (*Rauzi et al., 2015*). Our own observation in the *snail* mutant is also consistent with this result (*Figure 8e*). Interestingly, we found that in *snail* mutant embryos, cells moved toward both ventral and posterior sides as soon as ectodermal shortening began (*Figure 8d*). This is in sharp contrast to the wild-type embryos, where the onset of the posteriorly directed movement commenced 20–25 min later than the onset of $Vol_{ec}$ reduction and ventrally directed movement (*Figure 8c*). The altered pattern of cell movement in *snail* embryos could be explained by a combined effect of ectodermal compression and lack of ventral furrow invagination (*Figure 8f*). In this hypothetical scenario, the compression promotes ventral furrow invagination in wild-type embryos, which in turn functions as a 'sink' to facilitate (and thereby bias) the movement of the ectoderm in the ventral direction. However, since the *snail* mutant embryos do not form ventral furrow, an increase in ectodermal compression induces tissue movement in both AP and DV axis simultaneously, albeit slower.

So far, data from ours and others have shown that ventrally directed ectoderm movement still occurs without ventral furrow formation. It remains unclear, however, whether this decoupling of tissue movement can be observed without altering cell fate specification. Our optogenetic approach provided an opportunity for us to test this. First, we examined the ectodermal movement in the Early Group embryos immediately after stimulation. Our measurements confirmed the "reverse" movement of the ventral cells during tissue relaxation and demonstrated that during the same time period the lateral ectoderm moved in the opposite direction, toward the ventral midline (*Figure 9a–e*). Second,

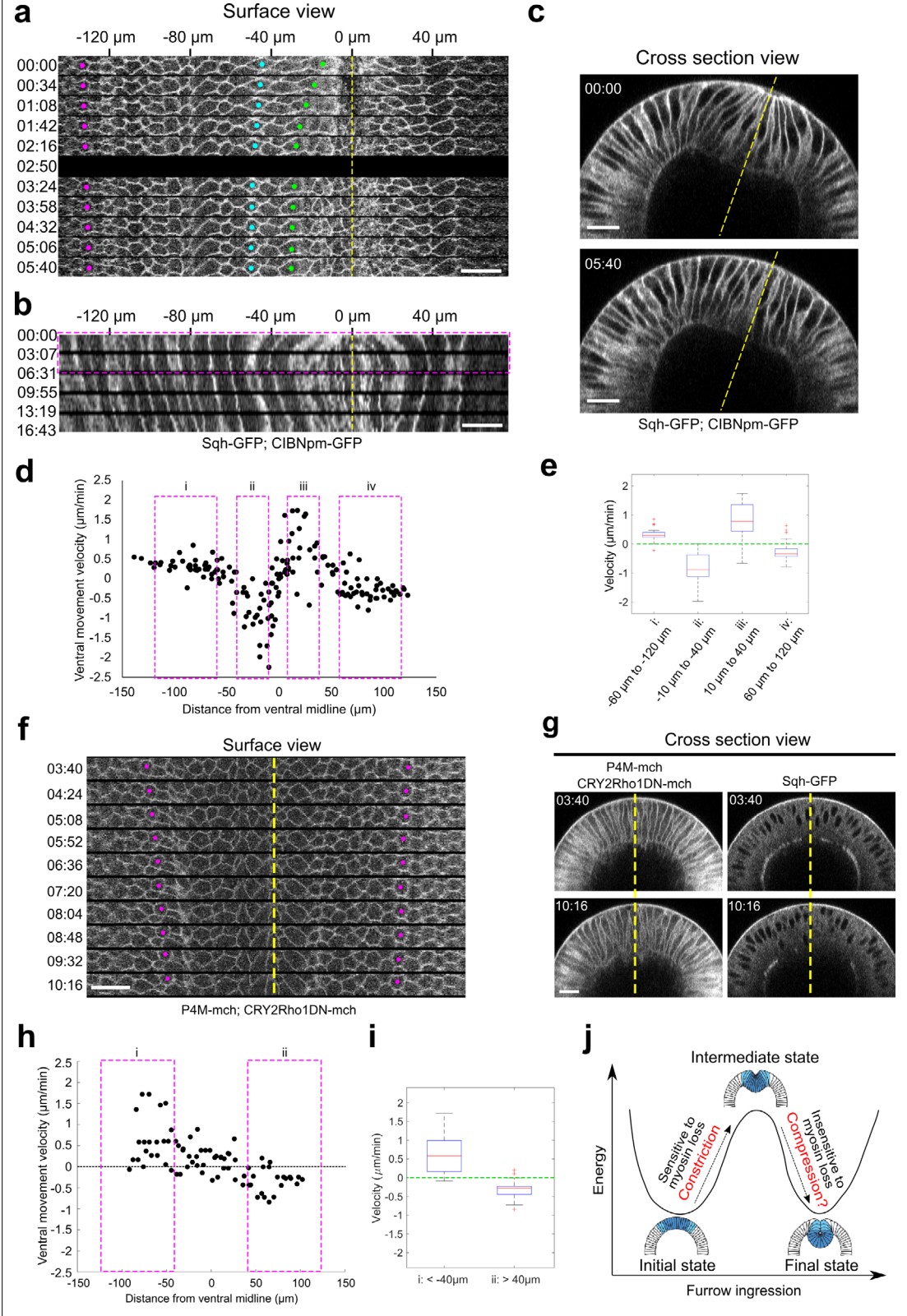

**Figure 9.** Lateral ectodermal cells still move ventrally in Early Group embryos when the ventral cells undergo apical relaxation upon Opto-Rho1DN stimulation. (**a**) Montage showing the surface view of an Early Group embryo during the first 6 min after stimulation (T=0). Yellow dashed line depicts ventral midline. Despite the tissue relaxation in the ventral region of the embryo (green and cyan dots), the ectodermal cells continue to move toward the ventral midline. (**b**) Kymograph generated from the surface view of the same movie as shown in (**a**). Magenta box indicates the time window where

*Figure 9 continued on next page*

*Figure 9 continued*

the average velocity of tissue movement is measured. (**c**) Cross-section view of the same embryo in (**a**) showing the relaxation of the apical indentation at the ventral side of the embryo. Yellow line: DV axis. (**d**) Velocity of tissue movement as a function of the initial distance from the ventral midline. N=7 Early Group embryos. (**e**) Box plot showing the velocity of tissue movement at ventral mesodermal ((ii): from –10 μm to –40 μm, and (iii): from 10 μm to 40 μm) or lateral ectodermal ((i): from –60 μm to –120 μm, and (iv): from 60 μm to 120 μm) regions of the embryo. Statistical analysis testing whether the measured velocities are significantly larger (i, iii) or smaller (ii, iv) than 0 is performed using one-tailed one-sample t-test against 0. p<0.001 for all tests. (**f, g**) Surface kymograph (**f**) and cross-section views (**g**) of embryos expressing Opto-Rho1DN and stimulated at the onset of apical constriction. Yellow line: ventral midline. Ectoderm cells (magenta dots) still move toward ventral midline even though no ventral furrow was formed (**g**). The left and right panels in (**g**) show the mCherry and GFP signals from the same embryo. N=3 embryos. T=0 is the onset of apical constriction. All scale bars: 20 μm. (**h**) Velocity of tissue movement as a function of the initial distance from the ventral midline. N = 3 *snail* mutant embryos. (**i**) Box plot showing the velocity of tissue movement at lateral ectodermal regions of the embryo ((i): from –40 μm to –110 μm, and (ii): from 40 μm to 110 μm). Statistical analysis testing whether the measured velocities are significantly larger (i) or smaller (ii) than 0 is performed using one-tailed one-sample t-test against 0. p<0.001 for both tests. (**j**) Cartoon depicting the proposed model for mechanical bistability of the mesoderm epithelium during gastrulation. Actomyosin-mediated apical constriction is important for the system to transition from the initial state to an intermediate state, whereas the subsequent transition into the final, fully invaginated state is facilitated by ectodermal compression and is insensitive to loss of actomyosin contractility. The presence and potential function of ectodermal compression remain to be tested experimentally.

The online version of this article includes the following source data for figure 9:

**Source data 1.** Measurements of velocity of tissue movement as a function of the initial distance from the ventral midline in Early Group embryos.

**Source data 2.** Measurements of velocity of tissue movement as a function of the initial distance from the ventral midline in Opto-Rho1DN embryos stimulated before gastrulation.

---

we inhibited Rho1 at the onset of apical constriction to mimic the ventral furrow phenotype of the *snail* mutants, but without changing mesodermal cell fate (*Figure 9f–i*). Despite the lack of apical constriction, we observed ventrally directed ectodermal movement similar to that in the *snail* mutants (*Figure 9f–i*). These results indicate that the ventrally directed lateral ectoderm movement can occur independent of ventral furrow formation when cell fate specification is expected to be normal. Taken together, our observations suggest that ectodermal shortening may serve as an independent mechanical input that could function together with other mechanical factors to facilitate mesoderm invagination. We propose that apicobasal cell shortening of the ectodermal cells (along with other unknown mechanisms) generates in-plane compression during gastrulation, which functions together with active cell shape changes in the mesoderm to facilitate mesoderm invagination through epithelial buckling (*Figure 9j*).

## Discussion

In this work, we developed an optogenetic tool, Opto-Rho1DN, to acutely inhibit actomyosin contractility in developing embryos. Using this tool, we discovered that while actomyosin-mediated active cell deformation is critical for mesoderm invagination, it is dispensable during the subsequent rapid invagination step after a transitional stage ($T_{trans}$), when the initial apical constriction mediated 'tissue priming' is completed. These observations suggest that the mesoderm epithelium is mechanically bistable during gastrulation—eliminating the active forces that deform the tissue at different stages of ventral furrow formation causes the tissue to either fall back to the initial stable state or progress to the final, fully invaginated state. Using computer modeling, we tested a possible mechanism based on the analogy of buckling of a compressed elastic beam induced by an indentation force. We demonstrate that the binary response of the tissue can be recapitulated in a simulated folding process jointly mediated by apical constriction in the mesoderm and apicobasal shrinkage of the ectoderm. In this simulation, apicobasal shrinking of the ectoderm is expected to generate in-plane compression and increase the propensity of the epithelium to buckle. Finally, we show that apicobasal shrinkage of lateral ectoderm occurs during gastrulation in both wild-type and *snail* mutant embryos and may thereby offer an independent mechanical input that functions together with other mechanical forces (such as apical constriction) to facilitate mesoderm invagination. We propose that ventral furrow formation is mediated through a joint action of multiple mechanical inputs. Apical constriction drives initial indentation of ventral furrow, which primes the tissue for folding, whereas the subsequent rapid folding of the furrow is promoted by the bistable characteristics of the mesoderm and by lateral myosin contractions in the constricting cells (*Figure 9j*). We further hypothesize that ectodermal compression induced by ectodermal shortening induces mechanical bistability in the mesoderm,

which in turn facilitates mesoderm invagination through a buckling-like mechanism (*Figure 9j*). Future experiments investigating the presence and potential function of ectodermal compression would be key to testing these hypotheses.

Our model proposes that compressive forces from the surrounding ectoderm contribute to mesoderm invagination and result in the binary tissue response upon acute actomyosin inhibition. The processes that may potentially generate ectodermal compression during mesoderm invagination remain to be determined. One possible candidate is germband extension. During germband extension, the convergence of tissue in the DV axis may provide a pushing force on the ventral mesodermal cells and lead to the binary tissue response to acute Rho1 inhibition. However, we do not favor this model for the following reasons. Since germ band extension is a process of convergent extension, the cellular flow along the DV axis (the 'convergence') is expected to be associated with an accompanying cellular flow along the AP axis (the 'extension'). While the DV movement of the ectoderm occurs at about the same time as the onset of rapid ventral furrow invagination, the AP movement of the ectoderm did not start until ~10 min later, when the ventral furrow has nearly fully invaginated (*Figure 8c*). These results are consistent with previous reports (*Lye et al., 2015*) and suggest that the initial DV movement is caused by the ventral furrow pull, rather than the convergence of the germband. Along the same vein, the apicobasal shortening of the lateral ectoderm also happens earlier than the onset of AP movement (*Figure 8c*), suggesting that germband extension does not account for ectodermal cell shortening during ventral furrow formation. Furthermore, in Late Group embryos, the cellular flow in the lateral ectodermal region displays little AP tissue movement during the course of ventral furrow invagination (*Figure 6—figure supplement 2*). Taken together, these observations suggest that germband extension is unlikely the cause of the binary tissue response observed in our optogenetic experiment. That being said, our results do not rule out the possibility that germband extension may contribute to the closing of ventral furrow at the final stage of furrow invagination. In addition, while we consider mechanical bistability as the most parsimonious explanation of the observed binary tissue response, we could not formally rule out other possible mechanisms.

In principle, bending of a cell sheet can be achieved through two general mechanisms. In the first mechanism, an active cell deformation (e.g., apical constriction) that results in cell wedging can cause the tissue to acquire a folded configuration. In this case, the mechanism is tissue autonomous and does not necessarily require active mechanical contributions from neighboring tissues. In the second mechanism, bending of a cell sheet is prompted by an increase in the compressive stresses within the tissue (*Mao and Baum, 2015*). This mechanism is tissue non-autonomous and depends on the tissues surrounding the region where folding happens. The canonical model for apical constriction-mediated cell sheet folding emphasizes the role of active cell deformation in the constriction domain (*Wolpert et al., 2015*). A number of recent studies, however, have implicated additional mechanical contributions from outside of the constriction domain that can influence apical constriction-mediated tissue folding (*Rauzi et al., 2015*; *Perez-Mockus et al., 2017*). Our results further support this notion and suggest that ventral furrow formation involves both tissue autonomous and non-autonomous mechanisms. In our proposed model, the generation of ectodermal compression increases the buckling propensity of the epithelium, whereas the active cell shape change induced by apical constriction triggers buckling at a defined position. Without apical constriction, ectodermal compression alone is not sufficient in triggering epithelial buckling due to the high energy barrier associated with the transition from the initial state to the final, fully invaginated state (e.g., in *snail* mutant or Opto-Rho1DN embryos stimulated prior to $T_{trans}$). The active cell shape change induced by apical constriction allows the tissue to overcome this energy barrier. The proposed requirement for this tissue level cooperation may explain the recent data showing that ectopic activation of Rho1 activity at different regions of cellular blastoderm in *Drosophila* results in different final furrow morphologies (*Rich et al., 2020*).

*Drosophila* ventral furrow formation is a robust process in that it can still happen in various scenarios when apical constriction is weakened (*Leptin and Grunewald, 1990*; *Parks and Wieschaus, 1991*; *Kölsch et al., 2007*; *Yevick et al., 2019*). Recent studies have revealed that the supracellular actomyosin network is intrinsically robust, with build-in redundancies to secure global connectivity of the network (*Yevick et al., 2019*). We propose that mechanical bistability of the mesoderm enabled by ectodermal shortening provides an alternative, tissue-extrinsic mechanism that allows proper invagination of the mesoderm when actomyosin contractility in the mesodermal cells is partially impaired due to genetic and environmental variations.

The use of Opto-Rho1DN in our work has revealed an unexpected, tissue-specific involvement of Rho1 in regulating cortical F-actin. We show that lateral F-actin in the ectoderm is resistant to acute Rho1 inhibition, whereas both medioapical and lateral F-actin in the mesoderm are more sensitive. The reason for the tissue-specific response of F-actin to Rho1 inhibition is not entirely clear. Previous studies suggest that the Rho1 effector Diaphanous (Dia), a formin protein, is responsible for the assembly of medioapical F-actin in the mesoderm during apical constriction (*Homem and Peifer, 2008*; *Mason et al., 2013*). This notion is consistent with our observation that medioapical actin rapidly disappeared after Rho1 inhibition. In the ectoderm, F-actin appears to be mainly distributed along the lateral membrane. This is in contrast to active Rho1, which has been found to be enriched at the junctional regions but not along the lateral membrane in ectodermal cells (*Garcia De Las Bayonas et al., 2019*). This observation is consistent with our data that acute Rho1 inhibition has a negligible impact on the lateral F-actin in the ectoderm and suggests the involvement of actin regulators other than the Rho1-Dia pathway (*Grevengoed et al., 2003*). Interestingly, it has recently been shown that the mesodermal cell fate determinant Twist and Snail function together to specify the spatial pattern of F-actin distribution in the mesoderm (*Denk-Lobnig et al., 2021*). The difference in the sensitivity of F-actin to acute Rho1 inhibition likely reflects cell type and subcellular location specific regulation of F-actin.

Actomyosin contractility provides a common force generation mechanism in development. To date, the studies of tissue mechanics have been largely focused on the mechanisms that generate contractile forces that actively deform the tissues (*Gilmour et al., 2017*; *Heisenberg and Bellaïche, 2013*; *Chanet and Martin, 2014*). It remains less well understood how the 'passive' mechanical status of the tissue influences the outcome of tissue deformation. The Opto-Rho1DN optogenetic tool developed in this work provides an effective approach to acutely disrupt the myosin-dependent force generation machinery, and therefore can help us begin to discern the role of 'active' forces and 'passive' mechanical properties in morphogenesis. Furthermore, our work highlights the importance of cooperation between 'local,' tissue-autonomous active force production and 'global,' tissue-nonautonomous mechanical contribution in tissue morphogenesis. It will be interesting to explore whether such cooperation plays a role in other cell sheet folding processes and whether mechanisms other than apical constriction can be employed to trigger folding in a compressed cell sheet.

## Materials and methods

**Key resources table**

| Reagent type (species) or resource | Designation | Source or reference | Identifiers | Additional information |
|---|---|---|---|---|
| Genetic reagent (*Drosophila melanogaster*) | Sqh-GFP | *Royou et al., 2002* | | |
| Genetic reagent (*D. melanogaster*) | Sqh-mCherry | *Martin et al., 2009* | | |
| Genetic reagent (*D. melanogaster*) | E-cadherin-GFP | *Oda and Tsukita, 2001* | | |
| Genetic reagent (*D. melanogaster*) | UASp-P4M-mCherry | *Chen and He, 2021* | | |
| Genetic reagent (*D. melanogaster*) | Utrophin-Venus | *Figard and Sokac, 2011* | | |
| Genetic reagent (*D. melanogaster*) | halo snail/CyO, Sqh-GFP | *Martin et al., 2009* | | |
| Genetic reagent (*D. melanogaster*) | UASp-CIBN-pm-GFP (II) | *Guglielmi et al., 2015* | | |
| Genetic reagent (*D. melanogaster*) | UASp-CIBN-pm (I) | *Guglielmi et al., 2015* | | |
| Genetic reagent (*D. melanogaster*) | UASp-CRY2-Rho1DN-mCherry (III) | This study | | See 'Fly stocks and genetics' for details |

*Continued on next page*

*Continued*

| Reagent type (species) or resource | Designation | Source or reference | Identifiers | Additional information |
|---|---|---|---|---|
| Genetic reagent (*D. melanogaster*) | *UASp-CIBN-pm-GFP; CRY2-Rho1DN* | This study | | See 'Fly stocks and genetics' for details |
| Genetic reagent (*D. melanogaster*) | *UASp-CIBN-pm; CRY2-Rho1DN* | This study | | See 'Fly stocks and genetics' for details |
| Genetic reagent (*D. melanogaster*) | *Maternal-Tubulin-Gal4 (67)* | *Hunter and Wieschaus, 2000* | | *Maternal-Tubulin-Gal4* on Chromosome II |
| Genetic reagent (*D. melanogaster*) | *Maternal-Tubulin-Gal4 (15)* | *Hunter and Wieschaus, 2000* | | *Maternal-Tubulin-Gal4* on Chromosome III |
| Genetic reagent (*D. melanogaster*) | *67; 15* | *Hunter and Wieschaus, 2000* | | *Maternal-Tubulin-Gal4* on II and III |
| Genetic reagent (*D. melanogaster*) | *67 Sqh-mCherry; 15 Sqh-GFP* | This study | | See 'Fly stocks and genetics' for details |
| Genetic reagent (*D. melanogaster*) | *67 Sqh-mCherry; 15 E-cadherin-GFP* | This study | | See 'Fly stocks and genetics' for details |
| Genetic reagent (*D. melanogaster*) | *67 Sqh-GFP; 15* | This study | | See 'Fly stocks and genetics' for details |
| Genetic reagent (*D. melanogaster*) | *67 Sqh-GFP; 15 Sqh-GFP* | This study | | See 'Fly stocks and genetics' for details |
| Genetic reagent (*D. melanogaster*) | *67 Sqh-mCherry; 15 Utrophin-Venus* | This study | | See 'Fly stocks and genetics' for details |
| Genetic reagent (*D. melanogaster*) | *67 Sqh-GFP; 15 P4M-mCherry* | This study | | See 'Fly stocks and genetics' for details |
| Recombinant DNA reagent | pTiger | S. Ferguson, State University of New York at Fredonia | | Transformation vector containing the attB site and 14 upstream UAS sites |
| Recombinant DNA reagent | pTiger-CRY2-mCherry-Rho1DN | This study | | Transformation construct containing CRY2-mCherry-Rho1DN |
| Software, algorithm | MATLAB | MathWorks | | https://www.mathworks.com/?s_tid=gn_logo RRID:SCR_001622 |
| Software, algorithm | Fiji | ImageJ | | http://fiji.sc RRID:SCR_002285 |
| Software, algorithm | Embryo Development Geometry Explorer (EDGE) | *Gelbart et al., 2012* | | https://github.com/mgelbart/embryo-development-geometry-explorer, *Gelbart, 2018* |
| Software, algorithm | ilastik | *Berg et al., 2019* | | https://www.ilastik.org/ RRID:SCR_015246 |

## Generation of CRY2-Rho1DN construct

To generate the CRY2-mCherry-Rho1DN fusion gene, the Rho1 CDS bearing an ACT to AAT mutation and a TGC to TAC mutation (T19N and C189Y, respectively) was synthesized and inserted to the C-terminal end of pCRY2PHR-mCherryN1 (Addgene, deposited by Chandra Tucker) through Gibson assembly. A 12-aa flexible linker sequence (GGGGSGGGGSGG) was included between the mCherry and Rho1DN sequences. The resulting CRY2PHR-mCherry-Rho1N18Y189 fusion gene was subsequently inserted into pTiger, a transformation vector containing the attB site and 14 upstream UAS GAL4-binding sites (courtesy of S. Ferguson, State University of New York at Fredonia, Fredonia, NY). The resulting pTiger-CRY2-mCherry-Rho1DN construct was sent to Genetic Services for integration into the attP2 site using the phiC31 integrase system (*Groth et al., 2004*).

## Fly stocks and genetics

Fly lines containing the following fluorescent markers were used: *Sqh-GFP* (*Royou et al., 2002*), *Sqh-mCherry* (*Martin et al., 2009*), *E-cadherin-GFP* (*Oda and Tsukita, 2001*), *P4M-mCherry* (*Chen and*

*He, 2021*), and *Utrophin-Venus* (*Figard and Sokac, 2011*). To generate *snail* mutant embryos, we used the *halo snail/CyO, Sqh-GFP* line (*Martin et al., 2009*).

Fly lines containing *CIBN-pm-GFP (II)*, *CIBN-pm (I)* (*Guglielmi et al., 2015*), or *CRY2-Rho1DN (III)* (this study, note that *CRY2-Rho1DN* also contains the *mCherry* tag) were used to generate the following stocks: *CIBN-pm-GFP; CRY2-Rho1DN* and *CIBN-pm; CRY2-Rho1DN*. To generate embryos containing maternally deposited CIBN-pm-GFP and CRY2-Rho1DN proteins, *CIBN-pm-GFP; CRY2-Rho1DN* females were crossed to males from the *Maternal-Tubulin-Gal4* line 67.15 to generate *CIBN-pm-GFP/67; CRY2-Rho1DN/15* females ('67' and '15' refer to *Maternal-Tubulin-Gal4* on chromosomes II and III, respectively [*Hunter and Wieschaus, 2000*]). To generate embryos containing maternally deposited CIBN-pm (without GFP) and CRY2-Rho1DN proteins, *CIBN-pm; CRY2-Rho1DN* females were crossed to *67.15* males to generate *CIBN-pm/+; +/67; CRY2-Rho1DN/15* females. In both cases, embryos derived from the F1 females were used in the optogenetic experiments. The use of CIBN-pm without a GFP tag was to allow better visualization of GFP-tagged myosin regulatory light chain Sqh. The following *Maternal-Tubulin-Gal4* lines were used: (1) *67; 15*, (2) *67 Sqh-mCherry; 15 Sqh-GFP*, (3) *67 Sqh-mCherry; 15 E-cadherin-GFP*, (4) *67 Sqh-GFP; 15*, (5) *67 Sqh-GFP; 15 Sqh-GFP*, (6) *67 Sqh-mCherry; 15 Utrophin-Venus*, and (7) *67 Sqh-GFP; 15 P4M-mCherry*. Specifically:

- *Figure 1c, d and g*, *Figure 1—figure supplement 1*: *CIBN-pm-GFP; CRY2-Rho1DN* females were crossed to *67; 15* males.
- *Figure 1f*: *CIBN-pm; CRY2-Rho1DN* females were crossed to *67 Sqh-GFP; 15* males.
- *Figure 2a*: *CIBN-pm* females were crossed to *67 Sqh-mCherry; 15 Utrophin-Venus* males.
- *Figure 2b*: *CIBN-pm; CRY2-Rho1DN* females were crossed to *67 Sqh-mCherry; 15 Utrophin-Venus* males.
- *Figure 3*: *CIBN-pm; CRY2-Rho1DN* females were crossed to *67 Sqh-mCherry; 15 E-cadherin-GFP* males.
- *Figures 4 and 5*, *Figure 4—figure supplements 1–2*, *Figure 5—figure supplement 1*, *Figure 6—figure supplements 1–2*, *Figure 8—figure supplements 1 and 3*, *Figure 9a–e*: *CIBN-pm-GFP; CRY2-Rho1DN* females were crossed to *67 Sqh-mCherry; 15 Sqh-GFP* males.
- *Figure 4—figure supplement 3a*: *CIBN-pm; CRY2-Rho1DN* females were crossed to *67 Sqh-mCherry; 15 Sqh-GFP* males.
- *Figure 4—figure supplement 3c*: *CIBN-pm; CRY2-Rho1DN* females were crossed to *67 Sqh-GFP; 15 Sqh-GFP* males.
- *Figure 9f–i*: *CIBN-pm; CRY2-Rho1DN* females were crossed to *67 Sqh-GFP; 15 P4M-mCherry* males.

## Viability test for Opto-Rho1DN embryos

Embryo viability test was performed by hand selecting cellularizing embryos derived from *67/CIBN-pm-GFP; 15/CRY2-Rho1DN* female flies in a dark room where the only illuminating light is red light. The selected embryos were randomly divided into two groups and placed on two fresh apple juice plates. One plate was kept in the dark and the other was placed under a beam of white light with moderate intensity. Plates were kept at 18°C for more than 48 hr, after which the hatched and unhatched embryos were counted using a Nikon stereo microscope. Three independent trials were conducted. In total, 60 out of 70 stimulated embryos did not hatch whereas all 63 unstimulated embryos hatched.

## Live imaging and optogenetic stimulation

To prepare embryos for live imaging, embryos were manually staged and collected from apple juice plates, and dechorionated with ~40% bleach (i.e., ~3% sodium hypochlorite) in a dark room using an upright Nikon stereo microscope. In optogenetic experiments, embryos were protected from unwanted stimulation using an orange-red light filter placed on top of white light coming from the stereo microscope. After dechorionation, embryos were rinsed thoroughly with water, and transferred on a 35-mm glass-bottom dish (MatTek Corporation). Distilled water was then added to the dish well to completely cover the embryos. Live imaging was performed in water at room temperature with one of the following approaches.

To examine the plasma membrane recruitment of CRY2-Rho1DN and its impact on cortical association of myosin (*Figure 1c–f*), embryos were imaged using a Nikon inverted spinning disk confocal microscope equipped with the perfect focus system and Andor W1 dual camera, dual spinning disk module. A CFI Plan Apo Lambda 60×/1.40 WD 0.13-mm oil objective lens was used for imaging.

For membrane recruitment of CRY2-Rho1DN, time-lapse movies were taken for post-cellularization embryos at a single z-plane close to the apical side of the tissue at a frame rate of 0.46 s per frame. For cortical association of myosin, a small Z-stack (five slices, 0.5 µm step size) was taken at a rate of 3.46 s per stack. The image size is 858 by 1536 pixels with a lateral pixel size of 0.11 µm, which corresponds to a 94 µm by 169 µm region. For each embryo, pre-stimulation images were acquired using a 561-nm laser, which did not activate Opto-Rho1DN. Stimulation and post-stimulation imaging were performed simultaneously as we subsequently imaged the embryo with both 488 nm and 561 nm lasers in an alternating pattern using triggered excitation.

To examine the spatial confinement of stimulation (*Figure 1g*) and the effect of activation of Opto-Rho1DN on apical constriction during ventral furrow formation (*Figure 1—figure supplement 1*), embryos were imaged on a Leica SP5 confocal microscope with a 63×/1.3 NA glycerin-immersion objective lens. A 2× zoom was used. Twenty confocal z-sections with a step size of 1 µm were acquired every 13 s. The image size is 1024×512 pixels with a lateral pixel size of 240 nm. The total imaged volume is approximately 246×123×19 µm³. First, a single-time-point pre-stimulated image stack was acquired using a 561-nm laser. Next, stimulation was performed by taking a single-time-point image stack using both 488 nm and 561 nm lasers, which took 13 s. Stimulation was followed by post-stimulation acquisition with 561 nm laser for 20 time points (260 s). The stimulation and post-stimulation acquisition cycle were repeated until the end of the movie (i.e., stimulated for 13 s every 273 s).

To examine the rate of tissue recoil after laser ablation (*Figure 3*), embryos were imaged on an Olympus FVMPE-RS multiphoton system with a 25×/1.05 numerical aperture water immersion objective lens. For unstimulated embryos, a pre-ablation Z-stack was obtained using a 1040-nm laser to image a 512×100-pixel region (171×33 µm², 3× zoom) with a step size of 2 µm, which took approximately 16 s. Laser intensity increased linearly (4%–7%) from the surface of the embryo to 100-µm deep. This pre-ablation Z-stack was used to determine the stage of the embryo. Next, a 10-frame single Z-plane pre-ablation movie was obtained with a 1040-nm laser. A 512×512-pixel region (171×171 µm³, 3× zoom) was imaged, which took approximately 1 s per frame. Next, a 920-nm laser with 30% laser intensity was used to ablate a 3D region from immediately below the vitelline membrane to ~20-µm deep. This was achieved by taking a Z-stack with a step size of 1.5 µm. The ROI of the ablated region is ~3 µm along the AP axis. The purpose of targeting multiple Z-planes for ablation was to ensure that the very apical surface of the ventral cells was ablated. This was particularly important for the stimulated embryos since the ventral cells undergo rapid apical relaxation after Rho1 inhibition. Immediately after laser ablation, a 100-frame single Z-plane post-ablation movie was acquired using both 1040 nm and 920 nm lasers. The same ROI was imaged as the pre-ablation single Z-plane movie with identical image acquisition speed. For stimulated embryos, a stimulation procedure was included before laser ablation. First, a Z-stack was acquired to determine the stage of the embryo, as described above. Then, a 458-nm diode laser with 0.3% laser intensity was used to illuminate the whole embryo under 1× digital zoom for 12 s to activate the optogenetic module. A 3-min wait time was applied after stimulation to ensure the complete inactivation of myosin and disassembly of apical F-actin before laser ablation. Following this wait time, the same laser ablation procedure as used for the unstimulated embryos was applied (pre-ablation single Z-movie, laser ablation, and post-ablation single Z-movie), except that both 1040 nm and 920 nm lasers were used for acquiring the pre-ablation movie.

The analyses of F-actin (Utrophin-Venus) after Opto-Rho1DN activation (*Figure 2*) and the stage-specific effect of Opto-Rho1DN activation on ventral furrow invagination (*Figure 4*) were conducted using the following imaging protocol. Embryos were imaged on the Olympus FVMPE-RS multiphoton system with a 25×/1.05 numerical aperture water immersion objective lens. Pre-stimulation images were obtained by using a 1040-nm laser to excite a 512×100-pixel region (171×33 µm², 3× zoom). Continuous imaging was conducted for pre-stimulation imaging. For each time point, a 100-µm Z-stack with step size of 1 µm was obtained with linearly increased laser intensity (4%–7.5%) over Z, which took 34 s. Stimulation was performed at different stages of ventral furrow formation using a 458 nm single-photon diode laser by illuminating the whole embryo under 1× zoom for 12 s with 0.3% laser intensity. A wait period of ~20 s was imposed after stimulation for examining the stage-specific effect. Post-stimulation images were obtained using both 1040 nm and 920 nm lasers. Same setting as pre-stimulation imaging was applied, with an addition of a linear increase in 920-nm laser intensity (0.1%–0.5%) over Z. Stimulation was conducted manually every five stacks during post-stimulation

imaging (i.e., stimulation for 12 s every ~200 s) to ensure the sustained membrane recruitment of CRY2-Rho1DN.

To image wild-type and *snail* mutant embryos for the volume measurement in the lateral ecto-derm (Vol$_{ec}$, *Figure 8*), embryos were imaged on the Olympus FVMPE-RS multiphoton system with a 25×/1.05 numerical aperture water immersion objective lens. Z stacks of 120 μm with a step size of 1 μm was acquired continuously over a 512×150-pixel region (256×75 μm², 2× zoom). A 920-nm laser with a linear increase of laser intensity over Z (0.3%–2.5%) was used for imaging. The temporal resolution of the movie was ~54 s/stack. *halo snail* homozygous mutant embryos were derived from the *halo snail/CyO, Sqh-GFP* stock and were recognized based on the 'halo' phenotype during cellularization.

To examine ectodermal cell movement after Opto-Rho1DN activation (*Figure 9f–i*), embryos were imaged on the Olympus FVMPE-RS multiphoton system with a 25×/1.05 numerical aperture water immersion objective lens. A pre-stimulation Z stack was acquired using 1040-nm laser to illuminate an ROI (512×150 pixels, 171×50 μm², 3× zoom) with a step size of 1 μm for a total depth of 100 μm. The laser intensity increased linearly over Z (3%–7%). Next, whole embryo stimulation was achieved by illuminating the embryo with a 458-nm laser at 0.3% laser intensity for ~12 s under 1× zoom. Post-stimulation Z stacks were acquired using similar protocol as the pre-stimulation Z stack, except that both 920 nm and 1040 nm lasers were used for imaging. The intensity of the 920-nm laser also increased linearly over Z (0.5%–2.5%). The temporal resolution of the post-stimulation movie was ~41 s/stack. At the end of each experiment, a single frame image of the embryo was acquired at 1× zoom to record the AP orientation of the embryo.

## Image analysis and quantification

All image processing and analysis were processed using MATLAB (The MathWorks) and ImageJ (NIH).

To quantify the percent membrane recruitment of CRY2-Rho1DN after stimulation (*Figure 1d*), cell membrane was segmented based on the CIBN-pm-GFP signal using a custom MATLAB code. The resulting membrane mask was used to determine the signal intensity of CRY2-Rho1DN (mCherry tagged) along the cell membrane over time. The intensity of the CRY2-Rho1DN signal in the cytoplasm, which is determined as the average intensity in the cells before stimulation, was subtracted from the measured intensity. The resulting net membrane signal intensity was further normalized by scaling between 0 and 1.

To quantify the rate of tissue recoil after laser ablation (*Figure 3*), the width change of the ablated region along the AP axis was measured over time from the kymograph using ImageJ. The width change during the first 20 s after laser ablation (when the change over time was relatively linear) was reported to indicate the rate of tissue recoil.

The invagination depth D (i.e., the distance between the vitelline membrane and the apex of the ventral-most cell, e.g., *Figure 4b*) and the apical-basal length of the ventral cells over the course of ventral furrow formation (*Figure 4—figure supplement 1a,b*) were manually measured from the cross-section view of the embryos using ImageJ.

To categorize embryo responses to stage-specific Opto-Rho1DN stimulation, first, each post-stimulation movie was aligned to a representative control movie based on the intermediate furrow morphology at the time of stimulation. The purpose of this alignment was to minimize the impact of embryo-to-embryo variation in T$_{L-S trans}$ (*Figure 4—figure supplement 1d*). The resulting aligned time was used in the analysis because it provided a better measure of the stage of ventral furrow formation than the 'absolute' time defined by the onset of gastrulation in each embryo. Next, dD/dt immediately after stimulation is determined by a linear fitting of D over time in the first 4 min after stimulation (*Figure 4—figure supplement 1f*). Embryos with dD/dt<–0.3 μm/min are defined as Early Group. These embryos undergo tissue relaxation and result in decrease in D over time. Embryos with dD/dt between –0.3 μm/min and 0.3 μm/min are defined as Mid Group. These embryos do not undergo obvious tissue relaxation but display a temporary pause before they continue to invaginate. Embryos with dD/dt>0.3 μm/min are defined as Late Group. These embryos continue to invaginate despite the rapid inactivation of myosin. To quantify the impact of myosin inhibition on furrow invag-ination (*Figure 4d*), the delay time in furrow invagination (T$_{delay}$) was measured. T$_{delay}$ is defined as the difference in time for a given stimulated embryo to reach a D of 20 μm compared to a representative control embryo.

To examine the change of apical cell area of the flanking, non-constricting cells in stimulated Opto-Rho1DN embryos (*Figure 4—figure supplement 2a, b*), a custom MATLAB script was used to generate a flattened surface view of the embryo which accounts for the curvature of the embryo. The apical cell area was measured by manually tracking and outlining flanking cells followed by area measurement using ImageJ.

To segment the mesoderm cells (*Figure 5*), Carving procedure from ilastik (*Berg et al., 2019*) was used in combination with manual correction using ImageJ and MATLAB to segment and reconstitute the 3D shape of individual cells. Cell length was calculated by summing up the distance between neighboring centroids at consecutive z planes to account for the curvature of the cell. Apical surface area was calculated based on the intersection area between the curved apical surface of the tissue and the 3D object of the cell. This intersection area was determined using a custom MATLAB code. The lateral surface area was calculated by a subtraction of the apical and basal surface area from the overall cell surface area, which was calculated from the 3D object of the cell. The volume is calculated directly from the 3D object of the cell.

To evaluate the volume flux in the lateral ectoderm region (*Figure 8*), pixel segmentation procedure from ilastik (*Berg et al., 2019*) was used to segment a fixed lateral ectoderm region (ROI: from 60° to 120° away from the ventral midline and 75 μm long along the AP axis) in both WT and *snail* mutant embryos. Segmentation was performed based on Sqh-GFP signal which clearly defines the apical and basal boundaries of the tissue. The tissue was segmented as a continuum without segmentation of individual cells. The volume of the selected ROI (Vol$_{ex}$) at each time point was determined based on the segmented 3D tissue object. To measure the movement of lateral ectoderm during gastrulation, the position of a cell initially located 50° away from the ventral midline at the onset of gastrulation was manually tracked over time using ImageJ.

To measure ectodermal shortening from 2D cross-section view of the embryos (*Figure 8—figure supplements 1 and 3*), the thickness of the lateral ectoderm is reported as the cross-section area of the ectodermal cell layer between 60° and 90° from the DV axis, which provides a good readout of tissue volume redistribution in 3D. A custom MATLAB script was used to facilitate the measurement. Specifically, the apical surface of the tissue was determined by automatic segmentation of the vitelline membrane of the embryo. The resulting curve was subsequently fit into a circle (the outer membrane circle). The basal surface of the tissue was determined by automatic segmentation and tracking of the basal myosin signal, with necessary manual corrections when the basal signal became too dim to be segmented accurately. Tissue thickness was measured as the cross-section area of a section of the ectodermal tissue bounded by the outer membrane, the basal surface, and two radii of the outer membrane circle that are 60° and 90° away from the ventral midline, respectively.

To detect ectodermal cell movement in Early Group embryos after stimulation (*Figure 9a–e*) and in embryos stimulated at the onset of gastrulation (*Figure 9f–i*), kymographs were generated from the surface views with the CIBN-pmGFP signal or P4M-mcherry that marks the cell membrane. Individual membrane traces within the first 6 min after stimulation were manually tracked from the kymograph using the line selection tool in ImageJ. The slope of the lines was subsequently measured using ImageJ to calculate the rate of membrane movement, which was then plotted against the position of corresponding membrane relative to the ventral midline at the time of stimulation. The average velocity of tissue movement at ventral mesodermal (from –10 μm to –40 μm and from 10 μm to 40 μm) or lateral ectodermal (from –60 μm to –120 μm and from 60 to 120 μm for Early Group embryos or <–40 μm and >40 μm for embryos stimulated at the onset of gastrulation) regions of the embryos was calculated and tested using one-tailed one-sample t-test against 0.

## Energy minimization-based vertex model for ventral furrow formation (Polyakov et al. Model)

The 2D vertex model for ventral furrow formation, which considers the cross-section view of the embryo, was constructed as previously described (*Polyakov et al., 2014*). In brief, the model contains a ring of 80 cells that represent the cross-section of an actual embryo. The initial geometry of the cells in the model resembles that in the real embryo. The apical, basal, and lateral membranes of the cells are modeled as elastic springs that resist deformation, and the cells have a propensity to remain constant cell volume (area in 2D). In the model, apical constriction provides the sole driving force for tissue deformation, which is achieved by allowing the apex of the cells in the ventral region of the

embryo to shrink. The resulting morphological changes of the tissue are determined by the stiffness of the apical, lateral, and basal springs, cell volume conservation, and a stepwise reduction of basal spring stiffness (see below).

The energy equation that approximates the mechanical properties of the system is described by the following expression:

$$E = \sum_i \varphi_i \mu_i a_i^2 + \sum_i \left[ K_l \left( l_i - l_0 \right)^2 + K_b \left( b_i - b_0 \right)^2 + K_a \left( a_i - a_0 \right)^2 \right]$$
$$+ \sum_i C_{VOL} \left( V_i \right) + C_{YOLK} \left( V_{yolk} \right)$$

The first term in the equation stands for myosin-mediated apical constriction. The term $\varphi_i \mu_i$ defines the spatial distribution of myosin contractility. $\varphi_i$ equals to one for cells within the mesoderm domain (9 cells on each side of the ventral midline, a total of 18 cells) and 0 for the rest of cells, which ensures that only the mesodermal cells will constrict apically. $\mu_i$ is a Gaussian function that peaks at the ventral midline. Specifically,

$$\mu_i = \mu_0 e^{\frac{-(i - i_{mid})^2}{2\sigma^2}}$$

Here, $i_{mid}$ stands for the cell ID at the ventral midline, $\mu_0$ defines the strength of apical constriction, and $\sigma$ defines the width of the force distribution. Note that due to the Gaussian-shaped force distribution, only 12 cells at the ventral most region of the embryo will undergo apical constriction, similar to what occurs in the real embryo.

The second term in the equation stands for the elastic resistance of the membranes. $a_i$, $b_i$, and $l_i$ stand for the area (length in 2D) of the apical, basal, and lateral membrane of cell , respectively. $a_0$, $b_0$, and $l_0$ are the corresponding resting length, which are set to be their initial length. $k_a$, $k_b$, and $k_l$ are the spring constant of the apical, basal, and lateral springs, respectively.

The third and fourth terms, $C_{VOL}$ and $C_{YOLK}$, are constraint functions that describe volume conservation of the cells and the yolk, respectively. These two terms impose penalties when the volume deviates from the resting value. Specifically,

$$C_{VOL} \left( V_i \right) = K_v \left( V_i - V_0 \right)^2$$
$$C_{YOLK} \left( V_{yolk} \right) = K_Y \left( V_{yolk} - V_{0, \, yolk} \right)^2$$

Here, $V_i$ stands for the volume of cell . $V_0$ stands for the initial equilibrium volume of cell . Similarly, $V_{yolk}$ stands for the volume of the yolk. $V_{0,yolk}$ stands for the initial equilibrium volume of the yolk. $K_v$ and $K_{yolk}$ control the deviation of the cell volume and the yolk volume from the initial equilibrium values, respectively.

In the simulation, the basal rigidity of the epithelium ($K_b$) decreases adiabatically, allowing the model to transition through a series of intermediate equilibrium states defined by the specific $K_b$ values. These intermediate states recapitulate the lengthening-shortening dynamics of the constricting cells observed in the real embryos (*Polyakov et al., 2014*).

To produce an in-plane compressive stress in the ectodermal tissue, we set the resting length of the lateral springs ($l_0$) 20% shorter than their original length. Because of the cell volume conservation constraint, shortening of the ectodermal cells in the apical-basal direction will result in an expansion of cells in the orthogonal direction, which is the planar direction of the epithelium. This expansion provides a mechanism to produce in-plane compressive stress in the epithelial sheet. As shown in *Figure 8—figure supplement 2*, when the percent reduction of the ectodermal cell length is lowered from 20% to 10%, the binary response to acute actomyosin inhibition is still present, although the final depth of the furrow in the simulated Late Group embryo is reduced. Further lowering the percent reduction to 5% abolishes the binary response.

To simulate the optogenetic inhibition of myosin contractility in the model, we set the program such that the apical contractility term will be reduced to 0 at defined intermediate equilibrium state. This approach allows us to inhibit apical contractility at any intermediate furrow configuration specified by $K_b$.

In order to simulate the impact of impairing apical constriction on invagination, we modified our model in two different ways. First, to reduce the number of apically constricting cells, we decreased

the width of the Gaussian function that defines the spatial distribution of apical myosin contractility. Second, to recapitulate the weakened and uncoordinated apical constriction observed in certain apical constriction mutants, we inhibited apical constriction from every other cell within the ventral 18 cell region.

In order to account for the impact of optogenetic stimulation of Opto-Rho1DN on lateral myosin in the constricting cells, we implemented an active lateral constriction force along the lateral edges of the constricting cells, on top of the passive lateral restoration force described above. In such a scenario, the active and passive lateral forces worked in combination to mediate ventral cell shortening, but only the active force was sensitive to myosin inactivation. For each lateral cortex within the constriction domain, the active lateral constriction force is generated by a spring with a resting length of 0. The lateral shortening force in the constricting cells is determined by:

$$F_{shortening} = K_{L\_active}l + K_{L\_passive}\left(l - l_0\right)$$

where $K_{L\_active}$ and $K_{L\_passive}$ are the spring constant of the active and passive lateral springs, respectively, and $l$ and $l_0$ are the current and the resting length of the lateral edges, respectively. As shown in **Figure 7**, a spring constant of $K_{L\_active} = 2$ generated the desired furrow morphology under a broad range of passive spring constant ($K_{L\_passive}$=[0.002, 20]). A further increase of $K_{L\_active}$ result in deeper furrows as more cells are incorporated into the neck region of the furrow. To account for the observation that lateral myosin in the constricting cells rapidly diminishes after Opto-Rho1DN activation, $K_{L\_active}$ is set to 0 after in silico myosin inhibition.

List of parameters used in **Figure 6c**:

| Parameter | Value |
|---|---|
| Ka | 30 |
| Kl | 20 |
| Kb* | $2^{10}$ to $2^0$ |
| μ0** | 5000 |
| σ | 3 |
| Kv | 5000 |
| KY | 1 |

*: In the simulation, $K_b$ decreases adiabatically from $2^{10}$ to $2^0$, with a twofold reduction at each step. An energy equilibrium state is reached for each value of $K_b$. These energy equilibrium states define the intermediate and final furrow morphologies.

**: The model recapitulates the binary response to acute loss of actomyosin contractility under a wide range of $\mu_0$ tested (500 – 50,000).

## Statistics

Sample sizes for the presented data and methods for statistical comparisons can be found in figure legends. p values were calculated using MATLAB ttest2 or rank-sum function.

## Acknowledgements

The authors thank members of the He lab and the Griffin lab at Dartmouth College for sharing valuable thoughts during this work; James Moseley and Magdalena Bezanilla for providing constructive suggestions and comments on the manuscript; Ann Lavanway for imaging support. The authors thank the Wieschaus lab and the De Renzis lab for sharing reagents, Oleg Polyakov for sharing the code for the energy minimization-based vertex model, and the Bloomington *Drosophila* Stock Center for fly stocks. This study is supported by NIGMS ESI-MIRA R35GM128745 and American Cancer Society Institutional Research Grant #IRG-82-003-33 to BH. The study used core services supported by STANTO15R0 (CFF RDP), P30-DK117469 (NIDDK P30/DartCF), and P20-GM113132 (bioMT COBRE).

## Additional information

### Funding

| Funder | Grant reference number | Author |
|---|---|---|
| National Institute of General Medical Sciences | ESI-MIRA R35GM128745 | Bing He |
| American Cancer Society | #IRG -82-003-33 | Bing He |
| Cystic Fibrosis Foundation | STANTO15R0 | Bing He |
| National Institute of Diabetes and Digestive and Kidney Diseases | P30-DK117469 | Bing He |
| Centers of Biomedical Research Excellence | P20-GM113132 | Bing He |

The funders had no role in study design, data collection and interpretation, or the decision to submit the work for publication.

### Author contributions

Hanqing Guo, Conceptualization, Data curation, Formal analysis, Investigation, Methodology, Software, Validation, Visualization, Writing - original draft, Writing - review and editing; Michael Swan, Methodology, Writing - review and editing; Bing He, Conceptualization, Data curation, Formal analysis, Funding acquisition, Investigation, Methodology, Project administration, Resources, Software, Supervision, Validation, Visualization, Writing - original draft, Writing - review and editing

### Author ORCIDs

Hanqing Guo http://orcid.org/0000-0001-8722-6253
Bing He http://orcid.org/0000-0002-8564-0933

### Decision letter and Author response

Decision letter https://doi.org/10.7554/eLife.69082.sa1
Author response https://doi.org/10.7554/eLife.69082.sa2

## Additional files

### Supplementary files

• Transparent reporting form

### Data availability

All data generated or analyzed during this study are included in the manuscript and supporting files. Source data files have been provided for the codes for the computer models described in this work and the numerical data for Figure 4 - figure supplement 1 and Figure 9.

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
