## [Editor Report]

The authors examine the process of mesoderm invagination in the *Drosphila* embryo and found that while myosin contractility is critical to prevent tissue relaxation during the early phase of the process, it is dispensable for the subsequent folding step. Through modeling and experimental analyses, the authors find that folding is likely mediated by a joint action of active cell shape changes in the mesoderm and apico-basal shrinking in the surrounding ectoderm and suggest that the mesoderm behave as a mechanically bistable tissue during gastrulation.

---

## [Decision Letter]

**Decision letter after peer review:**

Thank you for submitting your article "Mechanical bistability enabled by ectodermal compression facilitates *Drosophila* mesoderm invagination" for consideration by *eLife*. Your article has been reviewed by 3 peer reviewers, and the evaluation has been overseen by a Reviewing Editor and Utpal Banerjee as the Senior Editor. The following individuals involved in review of your submission have agreed to reveal their identity: Sebastian J Streichan (Reviewer #1); Magali Suzanne (Reviewer #3).

The authors address how contractile forces near the apical surface of a cell sheet drive out-of-plane bending of the sheet. To determine whether actomyosin contractility is required throughout the folding process and to identify potential actomyosin independent contributions for invagination, they develop an optogenetic-mediated inhibition of myosin and show that myosin contractility is critical to prevent tissue relaxation during the early stage of folding but is dispensable for the deepening of the invagination. The results shown in the first two figures support the idea that the mesoderm is mechanically bistable during gastrulation.

In the second part of this study, the authors test the role of the coupling between mesoderm and ectoderm by using 2D computational modelling and infrared pulsed laser dissection. They propose that the ectoderm can generate compressive forces on the mesoderm facilitating mesoderm internalization (2nd phase).

They then propose that this mechanical bistability arises from an in-plane compression from the surrounding ectoderm and that mesoderm invagination is achieved through the combination of apical constriction and tissue compression.

While the optogenetic experiments require additional controls, the overall results are compelling and deemed both interesting and significant for the field. By contrast, figure 4,5,6 appear highly speculative, and have substantial issues (e.g. reporting effects orders of magnitude below diffraction limit).

The manuscript presents two different models for data interpretation. The first one is a modified version of an earlier model that provides some predictions that can be tested with relatively simple experiments. On the other hand, the second model is rather complex and should be further analyzed with great care, before considering it for publication. It appears highly overparameterized, oftentimes using ad-hoc modifications for generating a desired effect. Moreover, it is a well-known fact from thin sheet elasticity that contributions of bending to total elastic energy are weighted by thickness cubed. The cell thickness shown are considerably thinner than the equivalent of cells in the embryo. At a thickness comparable to embryonic cells, bending will become orders of magnitude more costly. It further remains unclear how a dynamic variable is obtained. Thus, it is also unclear how the simulations ensure a robust trajectory in a high dimensional phase space with likely multiple minima.

After vigorous discussion with all reviewers, there emerged the possibility to focus the present manuscript on the original optogenetics findings, described in figure 1 and 2, and then quantitative analysis of predictions made by model shown in figure 3. These tests should include lateral edge lengths, across all cells in the ectoderm. Here it will be important to distinguish passive effects in the ectoderm due to pulling from the ventral furrow: If the furrow pulls, cells might actually also shorten laterally. This can be tested using wide-spread twist mutants. Finally, the authors need down their claim of compression. In the discussion, it may be mentioned the possibility of compression. However, the existing data does not support for such a mechanism at all.

Essential revisions:

1. Provide quantitative data of cell shape changes near the ventral furrow. Analysis should include both apical as well as lateral cell surface areas.

2. The authors analyze the effects of RhoDN on MyoII but never on the F-actin network. Rho1 is known to control F-actin organization so this should also be tackled thoroughly.

3. Test actomyosin contractility by measuring network recoil after laser dissection in control (RhoDN non activated) and RhoDN activated embryo.

4. Test the modification of the Polyakov model using available data. Since the lateral rest lengths are modified such that cells shorten over time (by 20 % – in real cells this would be about 8µm), if all 60 ectoderm cells shrink that much, this can result in a considerable in-plane expansion, assuming volume conservation. This could be tested by measuring the time course of average lateral length change of cells in the ectoderm (on the dorsal pole, and in the lateral regions), and explain how this compares to model assumptions.

5. Some degree of lateral cell shrinking is expected from ventral furrow pull. To distinguish possible contributions from ventral furrow vs active processes shortening the cell edges as proposed in the model, the authors should repeat the lateral cell surface analysis from (5) in twist or snail mutants.

6. To address to some extent the role of the mesoderm the authors could perform an early optogenetic RhoDN of the ventral side. If the ectoderm is pushing, then one could predict that the ventral cells should reduce their size along DV but not along AP because of the DV pushing from left and right ectoderms.

These experiments and the contrast of data with the modified Polyakov model may allow the authors to arrive at a soft conclusion implicating other forces, e.g. ectoderm compression, in the discussion.

[Editors' note: further revisions were suggested prior to acceptance, as described below.]

Thank you for resubmitting your work entitled "Mechanical bistability of the mesoderm epithelium facilitates mesoderm invagination during *Drosophila* gastrulation" for further consideration by *eLife*. Your revised article has been reviewed by 3 peer reviewers and the evaluation has been overseen by Utpal Banerjee as the Senior Editor, and a Reviewing Editor.

The manuscript has been improved but there are some remaining issues that need to be addressed, as outlined below:

Essential revisions:

The reviewers have found the manuscript much impproved and have praised the extensive work performed to address their criticism. This has resulted in a greatly improved manuscript that needs no further experimental work and is almost ready for publication pending some editorial changes. Specifically, they have pointed out one analysis that needs attention and several discussion points that should be addressed with text changes as follows:

1. For Figure 9f the authors have opted to track an ectoderm cell over time to demonstrate ectoderm displacement. This is fine, nevertheless the author should be consistent and perform again the analysis for Figure 9f by using the same analysis procedure they implemented for Figure 9a. More precisely the authors should follow a cell that is located 20 cells away from the midline (not just 10 cell away).

Discussion points:

2. The model should be described as a theory. In its current form it is hard to distinguish from the descriptions of experiments. It should be clearly labeled that it is purely elastic, and that it neglects the well-known viscous properties of tissues that dominate on the scale of at least 4 minutes and beyond.

3. On page 19 and eventually in the discussion the discrepancy between the model and the in vivo measurements should be discussed. More precisely, along a cross-section in the model the minimum necessary ectoderm cell shortening is 5-10% for 60 cells while in the real embryo the cell-shortening measured is 4-8% for a much smaller number of ectoderm cells closer to the mesoderm (20 cells?). Please consider these potential discrepancies and if they are indeed present discuss its possible origin and/or speculate in the discussion what might account for them.

4. In the new version of the paper, it is pointed out that F-actin in ectoderm cells is not affected by RhoDN optogenetic activation. This is quite puzzling and therefore merits at least further discussion.

5. In Figure 4A and in many other experiments, RhoDN is activated both in the mesoderm and in the ectoderm. Therefore, by following the logic of the model, ectoderm pushing is not dependent on Rho signaling. In other words, while mesoderm cell shortening depends on Rho, ectoderm cell shortening is Rho independent. This also is quite surprising thus merits further discussion.

6. The manuscript has valuable data on cell behaviors in the lateral ectoderm. But, the presentation is entirely focused around the idea of 'compression', and no alternatives discussed. One such alternative, could be the impact of germband extension on deepening the already formed furrow. The cellular flow of germband has a component directed towards the ventral pole, possibly allowing cells to flow into furrow, that has already formed. This extension will lead to the observed apicobasal shortening of lateral ectoderm cells, and deepening of the fold, but requires no mechanical bistablity. It would further be consistent with lack of fold and the described cell shapes at the apical surface in Snail and opto RhoDN experiment.

7. Stress is needed to build up for the proposed buckling by compression. However, the Snail experiments clearly demonstrate that no buckling of mesoderm occurs when cell behaviors in the mesoderm are perturbed. Stating mesoderm buckles due to compression from the ectoderm is, therefore, misleading and has not been demonstrated with an experiment. Any mention of this interpretation should be confined to the discussion.

8. In the discussion, the statement "Using computer modeling, we further demonstrated that mechanical bistablity in the mesoderm can arise from an apicobasal shrinkage of the ectoderm, which generates in-plane compression as the cell volume remains conserved" is misleading. This needs to be clarified, if the authors wish to raise the idea of bistability in the Discussion section.

9. While the authors consistently claim precise agreement between the model and data, it remains unclear to what extend this is the case. Visuals of simulations are provided. But there are no quantitative comparisons found that directly compare a model result with a corresponding measurement. Therefore, such claims (e.g. page 14 "In particular, the transitional state of the tissue revealed in the simulation is nearly identical to that identified in our ontogenetic experiments") have to be toned down (e.g. looks visually similar).

10. The quantitative analysis shown in figure 8 appears inconsistent with the descriptions. First, the authors refer to a rate of volume reduction, but show volume. Moreover, rate of volume reduction in WT appears consistently different from snail, yet is described as very similar. Such strong claims should either be toned down or backed up with a statistical significance test.

11. In the text page 20, the authors describe "In the wild type embryos, the compression promotes ventral furrow invagination, which in turn functions as a 'sink' to facilitate the movement of the ectoderm in the ventral direction". There is no experimental evidence provided for compression, and therefore this statement is speculation. Please rephrase.

*Reviewer #1 (Recommendations for the authors):*

The revised manuscript by Guo et al. has been revised, addressing some but not all of my concerns. In fact, the manuscript provides additional data, that argues strongly against the proposed mechanical bistability mechanism. The manuscript reads like two separate works.

The authors raise an interesting question in the abstract: Is myosin contractility at the apical surface required throughout folding? In the current version, characterization of the opto tool is much improved. It allows the authors to demonstrate apical myosin activity is not needed in a late phase of furrowing. The results presented in the first five figures are very interesting on their own, and provide a valuable contribution to the field. In my opinion, this would be an excellent point to stop the manuscript and enter the Discussion section. Such a work would be a great addition to *eLife*.

Instead, the authors enter a new direction, and propose mechanical bistablity of the mesoderm, to explain these ideas. This is somewhat unclear, as there are many possibly simpler explanations consistent with this very interesting observation that are not discussed (see more below). Instead, in figures six to nine, the manuscript hinges on speculation, and a purely elasticity-based model in combination with analysis of cell geometry along the apicobasal surface in support of their hypothesis. The conclusion, that invagination requires mechanical bistabilty of the mesoderm cannot be supported with the data presented. These claims should be toned down before publication, as already suggested in the first round of revisions. It seems this problem can be fixed by clarifications, and moving speculative data interpretation from the Results section to the Discussion sections.

(1) The manuscript offers no support for mechanical bistablity assumption. That purely elastic materials can buckle under compression is well established. But, the authors supply data that argues against mechanical bistablity in this system.

– Experimental data does not go beyond correlation, and yet the mechanism presented claims a causal role of ectoderm compression for ventral furrow folding. The original manuscript attempted to provide experimental data in support of the mechanical instability model. The current figures describe cell shapes in 3D, but there is no test of causality offered. This is somewhat puzzling, as the optogenetic tool should also function in the lateral ectoderm.

– The model should be described as a theory. In its current form it is hard to distinguish from the descriptions of experiments. It should be clearly labeled that it is purely elastic, and that it neglects the well-known viscous properties of tissues that dominate on the scale of at least 4 minutes and beyond.

– It is not clear to what extend the material in figure 8 supports the main argument. Instead, it seems to show the opposite. Lateral ectoderm shorting happens wether or not ventral furrow forms. This is a clear demonstration that the proposed mechanical bistability assumption is not able to drive tissue folding. Instead, these results suggest that folding needs to occur through an independent mechanism. Deepening of the fold could be generated by another mechanism (see below).

– Compression, as indicated by the authors, implies reduction in apical surface area of compressed cells, which is not shown in Snail or early opto RhoDN experiment. Early opto RhoDN experiments are described as heterogeneous cell morphology, but not further analyzed because of technical challenges. It is not clear how this is a technical problem, and not an issue of data interpretation. Heterogeneous apical cell area is consistent with cell shear, but not with external compression.

2) Discussion of model limitations and alternative scenarios.

– I congratulate the authors on their observation in figure 4b. It seems reasonable to further analyze this interesting phenomenon, and study the possible impact of tissue tissue interactions. In doing so, the manuscript would benefit from an open approach.

– The manuscript has valuable data on cell behaviors in the lateral ectoderm. But, the presentation is entirely focused around the idea of 'compression', and no alternatives discussed.

– One such alternative, could be the impact of germband extension on deepening the already formed furrow. The cellular flow of germband has a component directed towards the ventral pole, possibly allowing cells to flow into furrow, that has already formed. This extension will lead to the observed apicobasal shortening of lateral ectoderm cells, and deepening of the fold, but requires no mechanical bistablity. It would further be consistent with lack of fold and the described cell shapes at the apical surface in Snail and opto RhoDN experiment.

– The timescale of elasticity is very short compared to the 20 minutes of ventral furrow. As pointed out by the authors, the cited paper by Doubrovinsky provides an estimate for the transition to viscosity within 4 minutes. It is one of the longest currently published timescales for this process. As the authors clearly demonstrated turnover of the actomyosin cytoskeleton is very fast, further indicating the four minutes estimate is an upper bound of what is to be expected for these cells.

But, even if it is as long as four minutes, viscosity means, stresses will dissipate. Stress however is needed to build up for the proposed buckling by compression. Snail experiments clearly demonstrate that no buckling of mesoderm occurs when cell behaviors in the mesoderm are perturbed. Stating mesoderm buckles due to compression from the ectoderm is misleading, and has not been demonstrated with an experiment. Any mention of this interpretation should be confined to the discussion.

– In the discussion, the statement "Using computer modeling, we further demonstrated that mechanical bistablity in the mesoderm can arise from an apicobasal shrinkage of the ectoderm, which generates in-plane compression as the cell volume remains conserved" is misleading. The authors neither showed that ectoderm compresses mesoderm, nor is it a novel result that elastic systems can buckle under compression. This needs to be clarified, if the authors wish to raise the idea of bistability in the Discussion section.

– While the authors consistently claim precise agreement between the model and data, it remains unclear to what extend this is the case. Visuals of simulations are provided. But there are no quantitative comparisons found that directly compare a model result with a corresponding measurement. Therefore, such claims (e.g. page 14 "In particular, the transitional state of the tissue revealed in the simulation is nearly identical to that identified in our ontogenetic experiments") have to be toned down (e.g. looks visually similar).

*Reviewer #2 (Recommendations for the authors):*

Guo et al. have revised their paper by following the reviewer's suggestion.

The science presented is now more solid and merits publications after addressing the following 4 points:

1) In the new version of the paper, Guo and colleagues point out the fact that F-actin in ectoderm cells is not affected by RhoDN optogenetic activation. This is quite puzzling and therefore merits at least further discussion.

2) In Figure 4A and in many other experiments, RhoDN is activated both in the mesoderm and in the ectoderm. Therefore, by following the logic of the authors model, ectoderm pushing is not dependent on Rho signaling. In other terms, while mesoderm cell shortening depends on Rho, ectoderm cell shortening is Rho independent. This also is quite surprising thus merits further discussion.

3) For Figure 9f the authors have opted to track an ectoderm cell over time to demonstrate ectoderm displacement. This is fine, nevertheless the authors should be consistent and perform again the analysis for Figure 9f by using the same analysis protocol implemented for Figure 9a. More precisely the authors should follow a cell that is located 20 cells away from the midline (not just 10 cells away).

4) At page 19 and eventually in the discussion the authors should emphasize the discrepancy between the model and the in vivo measurements. More precisely they should make clear that along a cross-section in the model the minimum necessary ectoderm cell shortening is 5-10% for 60 cells while in the real embryo the cell-shortening measured is 4-8% for a much smaller number of ectoderm cells closer to the mesoderm (20 cells?).

*Reviewer #3 (Recommendations for the authors):*

The authors have done a great job responding to the concerns raised by the 3 referees with the quantitative analysis of cell shape after Rho1 inhibition, the analysis of the impact of Rho1 inhibition on F-actin, new laser ablation experiments to confirm the inactivation of myosin with their opto-Rho1DN construct, the analysis of the extend of endodermal shortening both on control and snail mutant embryos, the addition of active lateral contraction in the mesoderm in the model. They have added new data and analysis that strengthen the overall impact of the paper. They further discuss their findings in a more general context regarding previous works. I strongly support publication.

---

## [Author Response]

[…] Essential revisions:1. Provide quantitative data of cell shape changes near the ventral furrow. Analysis should include both apical as well as lateral cell surface areas.

We thank the reviewers for this suggestion and agree that a quantitative analysis of cell shape change after Rho1 inhibition will provide useful insights on the cellular basis of the different tissue behavior between the Early and Late Groups embryos. We performed three-dimensional reconstruction of a single row of cells along the medial-lateral axis from the ventral midline in three Early Group and three Late Group embryos. Because of the left-right symmetry of the embryo at this stage, we focused on the cells on one side of the ventral midline. The row of cells included the ventral mesodermal cells (~cell 1 – 6, “constricting cells”), the lateral mesodermal cells (~cell 7 – 9, “non-constricting flanking cells”), and presumably some ectodermal cells (~cell 10 – 11). The results of this analysis are now presented in Figure 5.

The major finding of this analysis is the differences in the cell behavior that explains the different tissue behavior in Early and Late Group embryos. For both group of embryos, the stimulation of Opto-Rho1DN resulted in an apical area relaxation in the constricting cells and an accompanying apical area reduction in the flanking cells (Figure 5). However, the apical area change results in very different change in cell curvature along the apical-basal axis. Before stimulation, cells located at the side of the constriction domain (cells 4 – 6, “side constricting cells”) and the flanking cells in both Early and Late Group embryos were bent over towards the ventral midline. After stimulation, all cells in Early Group embryos straightened up and partially restored their initial, columnar cell shape (Figure 5e, illustrated in Figure 5b). This cell shape change is associated with a reduction in the apical-basal cell length and lateral cell area (Figure 5d, i; Figure 5 – figure supplement 1; cells 4 – 9). In contrast, in Late Group embryos, only the flanking cell adjacent to the ectoderm (Figure 5j, cell 9) straightened up, whereas the side constricting cells and their neighboring flanking cells (cell 4 – 8) either remained their curvature or bent further towards the ventral midline (Figure 5j, illustrated in Figure 5g). A second major difference is that in Late Group embryos, the middle constricting cells (cell 1 – 3) underwent rapid cell shortening as the furrow continues to invaginate after stimulation (Figure 5i, illustrated in Figure 5g), which were not observed in Early Group embryos but rather resembles the normal cell shortening process during furrow invagination (Polyakov et al., 2014). Together, the continued bending of the side-constricting cells and the neighboring flanking cells and the continued shortening of the mid-constricting cells in Late Group embryos elucidates the cellular basis for the continued deepening of the furrow after Opto-Rho1DN stimulation and explains the tissue level difference between Early and Late Group embryos.

2. The authors analyze the effects of RhoDN on MyoII but never on the F-actin network. Rho1 is known to control F-actin organization so this should also be tackled thoroughly.

We thank the reviewers for raising this important question. During the revision, we examined the localization of an F-actin marker UtrophinABD-Venus before and after Opto-Rho1DN stimulation. The new data is now included in Figure 2.

In short, we found that stimulation of Opto-Rho1DN results in rapid diminishing of apical F-actin in the constricting cells. In unstimulated embryo, F-actin was enriched at the apical domain of the constricting cells (Figure 2a, cyan arrows). In addition, F-actin was also localized along the lateral membrane in both constricting and non-constricting cells (referred to as “lateral F-actin”, Figure 2a, yellow and magenta arrows). When we stimulated embryos during apical constriction, apical F-actin disappeared within 1.2 minutes after Rho1 inhibition (Figure 2b). The lateral F-actin in the constricting cells was not immediately affected and only appeared to diminish 4 minutes after stimulation (Figure 2b, T = 10.8 min, yellow arrows). In contrast, the lateral F-actin in the ectodermal cells was not significantly affected within the time frame of the experiment (Figure 2b, magenta arrows). Thus, stimulation of Opto-Rho1DN results in rapid inactivation of both apical myosin and diminishing of apical F-actin, both contributing to the rapid inactivation of apical actomyosin contractility.

3. Test actomyosin contractility by measuring network recoil after laser dissection in control (RhoDN non activated) and RhoDN activated embryo.

Following the reviewers’ suggestion, we measured tissue recoil along A-P axis after laser ablation in embryos at the lengthening phase of ventral furrow formation with or without stimulation. In Opto-Rho1DN stimulated embryos, we performed laser ablation 3 min after photo-activation of Opto-Rho1DN to ensure the complete inactivation of apical myosin and disassembly of apical F-actin. The new data are now included in Figure 3. One challenge associated with this experiment is that due to apical relaxation after stimulation, the cell apex moved rapidly to a more apical Z position. This change, together with the disappearance of apical myosin, made it difficult to identify the exact apical surface of the tissue. In order to ensure that we ablate the apical surface of the cells in both unstimulated and stimulated embryos, we ablated a 3D region of the tissue starting from immediately below the vitelline membrane to ~20 μm deep (illustrated in Figure 3a). Ablation was performed using a 920 nm femtosecond laser, with the laser intensity carefully tuned to avoid any non-specific heat-induced tissue damage. Quantifications showed a clear reduction of tissue recoil in the stimulated embryos compared to the unstimulated embryos, suggesting that Opto-Rho1DN stimulation during apical constriction results in a rapid loss of apical tension (Figure 3b, c and d).

4. Test the modification of the Polyakov model using available data. Since the lateral rest lengths are modified such that cells shorten over time (by 20 % – in real cells this would be about 8µm), if all 60 ectoderm cells shrink that much, this can result in a considerable in-plane expansion, assuming volume conservation. This could be tested by measuring the time course of average lateral length change of cells in the ectoderm (on the dorsal pole, and in the lateral regions), and explain how this compares to model assumptions.

We appreciate the reviewers’ suggestion. To further investigate the extent of ectodermal shortening during gastrulation, we performed 3D tissue volume measurement in the lateral ectodermal region of the wildtype embryo (ROI: 60-120 degrees away from the ventral furrow midline and 75 μm long along the AP axis; Figure 8a). We used the term “Vol_ec_” to refer to the volume of this ROI. We focused our analysis within the lateral ectodermal region since a previous study shows that the dorsal ectoderm does not significantly contribute to ventral furrow formation(Rauzi et al., 2015). We reasoned that if the ectodermal cells undergo apical-basal shortening, there should be a net volume outflux from the ROI, causing a reduction in Vol_ec_ (Figure 8a). Since cell shortening may not be uniform across the tissue, the change in Vol_ec_ provides a less noisy measure of the change in average tissue thickness. To measure Vol_ec_, we performed 3D tissue segmentation in the lateral ectoderm region without segmenting individual cells (Methods).

This analysis led to the following major observations, which are presented in Figure 8. First, Vol_ec_ increased during the first ten minutes of ventral furrow formation (Figure 8c). This observation is consistent with the previous measurement of ectodermal cells during gastrulation (Brodland et al., 2010) and this increase in Vol_ec_ likely reflects residual cellularization. Second, in all cases examined, Vol_ec_ started to decrease approximately halfway through ventral furrow formation, but the exact time of this transition varied between embryos (11.1 ± 2.3 min, mean ± s.d., n = 3 embryos; Figure 8f, arrowheads). Finally, the percent reduction in Vol_ec_ from the peak volume to that at the end of ventral furrow formation ranged between 4 – 8%. These observations are consistent with an independent 2D measurement of the cross-section area (as a proxy for tissue thickness) of a lateral ectodermal region (~ 6% reduction; Figure 8 – figure supplement 1).

In our vertex model, ectodermal compression was generated by letting all ectodermal cells shorten by 20% (ΔL = 20%; Methods), which was greater than the range measured in the actual embryos. We therefore tested the impact of reducing the extent of ectoderm shortening in our model We found that a minimum of 5 – 10% shortening in all 60 ectodermal cells is critical for generating the binary response (Figure 8 – figure supplement 2). When the percent reduction of the ectodermal cell length is lowered from 20% to 10%, the binary response to acute actomyosin inhibition is still present, although the final depth of the furrow in the simulated Late Group embryo is reduced. Further lowering the percent reduction to 5% abolishes the binary response – in this case, the intermediate furrow always relaxes back to the surface of the embryo after myosin inhibition regardless of the stage of stimulation.

A number of factors may account for the difference between the model and the actual embryo. For example, we observed residual cellularization in the ectoderm during the first few minutes of gastrulation, but there is no cell growth in the model. While an apical-basal shrinking of the ectodermal cells can generate in-plane compression, the onset and the effect of this shrinking may be underestimated by volume or length measurement if the cells are growing at the same time (Figure 8c, first 10 min). In addition, other morphogenetic processes happening during gastrulation, such as the formation of cephalic furrow and posterior midgut, might also contribute to the generation of tissue compression (Rauzi et al., 2015). Finally, while there is good evidence suggesting that the elasticity assumption is a reasonable approximation for tissue properties during ventral furrow formation (see our response to Reviewer #1’s comment #1 for further elaboration on this point), the embryonic tissue is viscoelastic instead of purely elastic.

However, despite these limitations, the model successfully recapitulated several main cell morphological features during ventral furrow formation, such as cell lengthening during apical constriction and shortening/wedging during invagination. The model also correctly predicted the ability of the mesoderm tissue to invaginate when apical constriction is partially impaired. Finally, the model recapitulated nearly all aspect of tissue response to acute actomyosin inhibition, including the binary tissue response, the morphology of the intermediate furrow at the transitional state, and the reduced final furrow width but not furrow depth in Late Group embryos. Using a minimalistic approach, the model was designed to identify the most critical active and passive mechanical inputs that can lead to the observed tissue behavior. The agreement between many major experimental observations and the modeling predictions suggests that the model captures the key features of the mechanism underlying the process. That being said, we acknowledge that future research is required to further understand the behavior and function of the ectodermal tissue in mesoderm invagination.

5. Some degree of lateral cell shrinking is expected from ventral furrow pull. To distinguish possible contributions from ventral furrow vs active processes shortening the cell edges as proposed in the model, the authors should repeat the lateral cell surface analysis from (5) in twist or snail mutants.

It is indeed important to distinguish the active (e.g. “apical-basal contraction” of the lateral ectoderm) versus passive (e.g. ventral furrow pull) mechanisms that may account for lateral cell shrinking during ventral furrow formation. Following this suggestion, we repeated the 3D lateral volume (Vol_ec_) measurements in *snail* mutant embryos. The new results are included in Figure 8. We observed a similar reduction in Vol_ec_ in *snail* mutant embryos as in WT embryos. Specifically, the reduction of Vol_ec_ occurred at about the same time as in the wildtype. In addition, the rate of Vol_ec_ reduction was similar between the wildtype and *snail* mutant embryos, although the *snail* embryos on average had a smaller Vol_ec_ to start with (Figure 8d and f). This result indicates that pulling from ventral furrow cannot fully account for the lateral cell shortening in the wildtype embryos and suggests the presence of ventral furrow-independent mechanism for ectodermal shortening.

Interestingly, we also observed a different pattern of lateral ectodermal movement in *snail* mutant embryos when compared to the wildtype embryos. In wildtype embryos, the onset of Vol_ec_ reduction is in general aligned with the onset of ventrally directed cell movement, whereas the onset of the posteriorly directed movement happens 20 – 25 minutes earlier (Figure 8c). However, in *snail* mutant embryos, cells move towards both ventral and posterior sides as soon as ectodermal shortening begins (Figure 8d). This observation can be explained by the combined effect of ectodermal compression and lack of ventral furrow invagination (Figure 8f). In wildtype embryos, the compression promotes ventral furrow invagination, which in turn functions as a “sink” to facilitate (and thereby bias) the movement of the ectoderm in the ventral direction. However, since the *snail* mutant embryos do not form ventral furrow, an increase in ectodermal compression induces tissue movement in a both A-P and D-V axis simultaneously, albeit slower.

Taken together, our data show that both ectoderm shortening and the associated ventrally directed ectoderm movement still occur without ventral furrow formation. These observations suggest that ectoderm shortening may serve as an independent mechanical input that facilitates mesoderm invagination by promoting mechanical bistability in the mesoderm.

6. To address to some extent the role of the mesoderm the authors could perform an early optogenetic RhoDN of the ventral side. If the ectoderm is pushing, then one could predict that the ventral cells should reduce their size along DV but not along AP because of the DV pushing from left and right ectoderms.

Following the reviewers’ suggestion, we performed the early stimulation experiment and presented the new data in Figure 9. A minor deviation from the suggested experiment is that instead of measuring apical domain size/shape of the ventral cells, we directly measured the movement of the lateral ectodermal cells after stimulation. This is mainly due to technical reasons. After Rho1 inhibition and disappearance of apical myosin, the cell morphology became more heterogeneous over time, which complicated the evaluation of cell aspect ratio. We reasoned that if the ectoderm is pushing the mesoderm from both sides, we should be able to observe movement of the ectodermal cells towards the ventral region even if the ventral furrow is inhibited (like in the *snail* mutant). Our result is consistent with this prediction (Figure 9f-g). In addition, the result is also consistent with our observation in Group Early embryos (Figure 9d and 9f-g). As reported in our original manuscript, in Early Group embryos, despite the immediate relaxation of the constricting cells after stimulation, the lateral ectoderm continued to move towards the ventral midline during the same time period (Figure 9a-9e). Together, the results of these experiments indicate that the ventrally directed movement of lateral ectoderm can occur independent of ventral furrow formation. These observations suggest that ectodermal shortening may serve as an independent mechanical input that promotes mechanical bistability in the mesoderm and thereby facilitates mesoderm invagination.

These experiments and the contrast of data with the modified Polyakov model may allow the authors to arrive at a soft conclusion implicating other forces, e.g. ectoderm compression, in the discussion.

Additional modeling analysis to test the known mechanisms for mesoderm invagination:

In addition to the new experiments mentioned above, we also included additional modeling analysis to further test the known mechanisms that promote ventral furrow invagination (as suggested by Reviewer #3). In particular, we sought to further test the model in light of the recent findings that myosin accumulates along the lateral membrane of the constricting cells (“lateral myosin”) facilitates cell shortening and tissue invagination (Gracia et al., 2019; John and Rauzi, 2021). In our optogenetic experiments, we found that lateral myosin diminished rapidly after Opto-Rho1DN stimulation, but this phenomenon was not considered in our original vertex model.

In our vertex model presented in original Figure 3 (current Figure 6), shortening of the constricting cells is mediated by passive elastic restoration forces generated in these cells as their lateral edges are stretched during cell lengthening (Polyakov et al., 2014). This passive elastic force is not affected by in silico myosin inactivation. To account for our observations in the actual optogenetic experiments, we incorporated an active lateral constriction force in the constricting cells that is sensitive to in silico myosin inactivation. In such a scenario, the active and passive lateral forces worked in combination to mediate ventral cell shortening, but only the active force was sensitive to myosin inactivation (Methods). The active and passive lateral forces are given by *K_L_active_l* and *K_L_passive_(l-l_0_)*, respectively, where *l* and *l_0_* are the current and the resting length of the lateral edge, respectively. As expected, the addition of the active lateral force allows us to reduce the passive lateral force while still generate furrows with normal morphology (Figure 7a). This modification allowed us to examine how the binary response of our model would be influenced if myosin inactivation impairs both apical constriction and cell shortening in the constriction domain. The result of this new analysis is presented in Figure 7 and discussed in the main text.

In short, we tested the impact of myosin inhibition with the combinations of *K_L_active_* and *K_L_passive_* that give rise to normal furrow morphology (Figure 7a, green shaded region). We found that the binary response of the model to myosin inhibition still exists when the cell shortening forces in the mesoderm become sensitive to myosin inhibition. However, a minimal level of myosin-independent lateral elastic force (*K_L_passive_* > 0.2) is required for the model to generate realistic final furrow morphology in Late Group embryos (Figure 7b). These results predict that mesoderm invagination demands an adequate amount of cell shortening force in the constricting cells that is insensitive to Rho1 inhibition. Interestingly, we found that lateral F-actin in the constricting cells persisted for several minutes after Opto-Rho1DN stimulation (Figure 2b, yellow arrows), which might provide the lateral elasticity described in our model.

[Editors' note: further revisions were suggested prior to acceptance, as described below.]

Essential revisions:The reviewers have found the manuscript much improved and have praised the extensive work performed to address their criticism. This has resulted in a greatly improved manuscript that needs no further experimental work and is almost ready for publication pending some editorial changes. Specifically, they have pointed out one analysis that needs attention and several discussion points that should be addressed with text changes as follows:1. For Figure 9f the authors have opted to track an ectoderm cell over time to demonstrate ectoderm displacement. This is fine, nevertheless the author should be consistent and perform again the analysis for Figure 9f by using the same analysis procedure they implemented for Figure 9a. More precisely the authors should follow a cell that is located 20 cells away from the midline (not just 10 cell away).

We thank the reviewers for this suggestion. Following the suggestion, we performed the same quantification as in Figure 9a for the phenotype described in Figure 9f. The new results are now incorporated into the updated Figure 9 (Figure 9h and 9i). Although the data are a little noisy, there is a quasi-linear relationship between the velocity of cell movement and the initial distance of the cell from the ventral midline (Figure 9h). We further examined the cell velocity in the -110 μm to -40 μm region and the 40 μm to 110 μm region, respectively. These regions correspond to ~ 7 cells to ~14 cells away from the ventral midline at the beginning of gastrulation. Our result shows that the velocities of both groups of cells are significantly different from zero (Figure 9i). These quantifications are consistent with our visual inspection and confirm that ectoderm cells move towards ventral midline even when apical constriction is inhibited.

Discussion points:2. The model should be described as a theory. In its current form it is hard to distinguish from the descriptions of experiments. It should be clearly labeled that it is purely elastic, and that it neglects the well-known viscous properties of tissues that dominate on the scale of at least 4 minutes and beyond.

Following reviewer’s suggestion, in the revised manuscript, we carefully distinguished between descriptions of experimental observations and conclusions drawn from the modeling analysis. Since the computer model is chosen to test the ideas of compression and buckling-like behavior, we are not able to avoid mentioning these terms before the Discussion section; however, we make it explicit that these concepts are proposed model rather than experimental conclusions.

To emphasize the difference between the actual embryo mechanical properties and that of the model we used in our work, we added the following sentence to the revised manuscript to state the limitation of the model and the reason why we chose this model to test our hypothesis:

“Note that the elasticity assumption in this model is a simplification of the actual viscoelastic properties of the embryonic tissue. […] It is therefore advantageous to use this model to explore the main novel aspect of the folding mechanics underlying ventral furrow formation and to test the central concept of our hypothesis based on relatively small number of assumptions.” (Page 15).

3. On page 19 and eventually in the discussion the discrepancy between the model and the in vivo measurements should be discussed. More precisely, along a cross-section in the model the minimum necessary ectoderm cell shortening is 5-10% for 60 cells while in the real embryo the cell-shortening measured is 4-8% for a much smaller number of ectoderm cells closer to the mesoderm (20 cells?). Please consider these potential discrepancies and if they are indeed present discuss its possible origin and/or speculate in the discussion what might account for them.

There is indeed a difference between the model and the in vivo measurements regarding the extent of ectodermal shortening that is needed in order to generate the binary tissue response in the model versus the actual extent of ectodermal shortening observed in the embryo. In the previous version of the manuscript (page 19), we discussed these discrepancies as follows:

“In our vertex model, ectodermal compression was generated by letting all ectodermal cells shorten by 20% (ΔL = 20%; Methods), which was greater than the range measured in the actual embryos. […] While our model recapitulated the main cell morphological features during ventral furrow formation and correctly predicted the binary response to acute actomyosin inhibition, future research is needed to elucidate the mechanism and function of ectodermal shortening in gastrulation.”

In the revised manuscript, we further expanded the discussion on this matter, as shown below:

“In our vertex model, ectodermal compression was generated by letting all ectodermal cells shorten by 20% (ΔL = 20%; Methods). […] While our model recapitulated the major cell morphological features during ventral furrow formation and correctly predicted the binary response to acute actomyosin inhibition, future research is needed to elucidate the mechanism and function of ectodermal shortening in gastrulation.” (Page 20)

4. In the new version of the paper, it is pointed out that F-actin in ectoderm cells is not affected by RhoDN optogenetic activation. This is quite puzzling and therefore merits at least further discussion.

This is a very interesting question. Our Rho1 inhibition showed that F-actin in the ectoderm is resistant to acute Rho1 inhibition, whereas the F-actin in the mesoderm is more sensitive. Specifically, apical F-actin rapidly disappeared within 1.2 minutes after Rho1 inhibition, whereas the lateral F-actin appeared to diminish 4 minutes after stimulation in the mesoderm. The reason for the tissue-specific response of F-actin to Rho1 inhibition is not entirely clear. Previous studies suggest that the Rho1 effector Diaphanous (Dia), a formin protein, is responsible for the assembly of medioapical F-actin in the mesoderm during apical constriction (Homem and Peifer, 2008; Mason et al., 2013). This notion is consistent with our observation that medioapical actin rapidly disappeared after Rho1 inhibition. In the ectoderm, F-actin appears to be mainly distributed along the lateral membrane. While Rho1 protein is uniformly distributed along the apical-basal axis of the cell, active Rho1(as detected by GFP-tagged Rho1 binding domain of anillin) is enriched at the junctional regions (and to some extent medioapcial region), but not along the lateral membrane in ectodermal cells (Garcia De Las Bayonas et al., 2019). This observation is consistent with our data that acute Rho1 inhibition has a negligible impact on the lateral F-actin in the ectoderm and suggests the involvement of actin regulators other than the Rho1-Dia pathway (Grevengoed et al., 2003). Interestingly, it has recently been shown that the mesodermal cell fate determinant Twist and Snail function together to determine specify the spatial pattern of F-actin distribution in the mesoderm (Denk-Lobnig et al., 2021). The difference in the sensitivity of F-actin to acute Rho1 inhibition likely reflects cell type and subcellular location specific regulation of F-actin.

We have now included the above discussion in the revised Discussion (Page 27 – 28).

5. In Figure 4A and in many other experiments, RhoDN is activated both in the mesoderm and in the ectoderm. Therefore, by following the logic of the model, ectoderm pushing is not dependent on Rho signaling. In other words, while mesoderm cell shortening depends on Rho, ectoderm cell shortening is Rho independent. This also is quite surprising thus merits further discussion.

This is an excellent point. To gain further insight on this issue, we examined the potential impact of acute Rho1 inhibition on ectodermal cell length change. We measured the ectoderm cross-section area as a proxy of the ectodermal cell length for Late Group embryos (following the protocol described in Figure 8—figure supplement 1). The new data is presented in Page 21 – 22 and Figure 8—figure supplement 3 of the revised manuscript.

Interestingly, in three out of five Late Group embryos we examined, Opto-Rho1DN stimulation resulted in a slight increase in ectoderm thickness (Figure 8—figure supplement 3a, magenta boxes). This increase is both mild and transient, and the trend of change in thickness rapidly (in 2 – 3 min) resumed the pre-stimulation pattern. In the remaining two out of five Late Group embryos, Opto-Rho1DN stimulation did not have an obvious impact on ectoderm thickness (Figure 8—figure supplement 3a, blue boxes). To further analyze this observation, we compared the Late Group embryo to the non-stimulated control embryos for ectoderm area change within a 1.7 min duration, either immediately after stimulation (Late Group embryos) or after the maximal ectoderm area was reached (non-stimulated control embryos). In control embryos, the ectoderm area on average reduced by 1%, whereas in Late group embryos, the area on average increased 0.6% (Figure 8—figure supplement 3b).

Since the impact of Opto-Rho1DN stimulation on ectoderm thickness change was both mild and transient even though Rho1 inhibition is persistent (Methods), we do not favor the explanation that ectodermal shortening is directly influenced by Rho1 inhibition. Instead, we hypothesized that this impact was an indirect consequence of what happened near the constriction domain upon Rho1 inhibition. Interestingly, we have previously observed that Rho1 inhibition results in moderate but detectable relaxation of the flanking non-constricting cells in some Late Group embryos (cell 9 in original Figure 5f – j; original Figure 6—figure supplement 1). This mild and transient relaxation, although not sufficient to cause a pause in the invagination of the constricting cells (original Figure 5f), may result in a slight “ventral-to lateral” flow of cell volume, thereby causing a temporary increase in ectodermal cell length. To test this possibility, we examined the correlation between the extent of ectoderm cell length increase and the reverse movement of flanking cells after Rho1 inhibition and detected a positive correlation (R^2^ = 0.667, n = 5 embryos, plot not shown). Although we were not able to draw solid conclusion on this correlation due to the small sample size, this result is consistent with the idea that Rho1 inhibition does not directly impact ectoderm shortening but transiently and mildly influences ectoderm cell length due to relaxation of cells adjacent to the constricting domain.

6. The manuscript has valuable data on cell behaviors in the lateral ectoderm. But, the presentation is entirely focused around the idea of 'compression', and no alternatives discussed. One such alternative, could be the impact of germband extension on deepening the already formed furrow. The cellular flow of germband has a component directed towards the ventral pole, possibly allowing cells to flow into furrow, that has already formed. This extension will lead to the observed apicobasal shortening of lateral ectoderm cells, and deepening of the fold, but requires no mechanical bistablity. It would further be consistent with lack of fold and the described cell shapes at the apical surface in Snail and opto RhoDN experiment.

We thank the reviewers for raising this very interesting possibility. However, based on previous literature and our own observations, we do not favor the model that germband extension contribute to the transition from apical constriction to invagination. Although there is a prominent DV-directed cellular flow of the lateral ectoderm during ventral furrow, this cellular flow could be due to ventral furrow pull or germ band extension, or both. Since germ band extension is a process of convergent extension, the cellular flow along the DV axis (the “convergence”) is expected to be associated with an accompanying cellular flow along the AP axis (the “extension”). To ask whether the DV tissue flow is due to germband extension or due to ventral furrow pull, we compared the timing of ventral furrow invagination and the ectoderm cell movement in the wildtype embryos by examining the data from original Figure 8. While the DV movement of the ectoderm occurred at about the same time as VF invagination, the AP movement of the ectoderm did not start until ~ 20 min after the onset of apical constriction, when the ventral furrow has nearly fully invaginated (with an invagination depth D larger than 40 μm; Figure 8c and 8f). For comparison, the transition phase (T_trans_) of VF formation is at about ~6 min after the onset of apical constriction, and the invagination depth D at T_trans_ is ~ 7 μm (Figure 4). These results are most consistent with a model that the DV movement of the ectoderm before T = 20 min is due to ventral furrow pull, instead of germ band extension. Along the same vein, a similar comparison between ectodermal shortening and ectodermal tissue flow indicates that ectodermal shortening occurs before the AP movement (Figure 8c), suggesting that germband extension does not account for ectodermal cell shortening before T = 20 min. These results are consistent with the notion in the literature that germband extension starts when ventral furrow formation is about to complete (Lye et al., 2015). Finally, we examined the cellular flow in Late Group embryos and observed little AP tissue movement during the course of VF invagination (Figure 6—figure supplement 2). Together, these results suggest that germband extension is unlikely to be the cause of the binary tissue response observed in our optogenetic experiment or the cause of ectodermal shortening before T = 20 min. That being said, our results do not rule out the possibility that germband extension may contribute to the closing of VF at the final stage of VF invagination, which will be an interesting topic for future research. In addition, while we consider mechanical bistability as the most parsimonious explanation of the observed binary tissue response, we could not formally rule out other possible mechanisms.

The above discussion about germband extension is included in the revised manuscript (Results: Page 15; Discussion: Page 25 – 26).

Regarding the comment on the *snail* mutant and early Opto-Rho1DN experiments:

The mechanical bistability model is not in odd with the lack of tissue folding in *snail* mutant and the early Opto-Rho1DN experiments. In our proposed model, ectodermal compression poises the mesoderm for invagination (i.e., transition from the initial stable equilibrium to the final, fully invaginated stable equilibrium); however, invagination does not happen instantaneously due to a high energy barrier between the two stable configurations. We propose that apical constriction provides the mechanism for the system to overcome this energy barrier and allows the tissue to invaginate. In this process, apical constriction is necessary to bring the system to a transitional state but is dispensable for the subsequent transition into the final fully invaginated state. In the *snail* mutant, the ventral epithelium is “stuck” at the initial configuration even in the presence of ectodermal compression due to the lack of mechanical input (“apical constriction”) to allow it to overcome the high energy barrier. Such a system is analogous to a bistable light switch. The switch lever has two stable positions (“on” and “off); however, the lever will not spontaneously switch between the two states unless there is an energy input from a push.

We have modified the text in the revised manuscript to make our points more explicit. In particular, we emphasize that in the model/mechanism we propose, apical constriction provides the essential trigger for mesoderm buckling in the presence of ectodermal compression.

7. Stress is needed to build up for the proposed buckling by compression. However, the Snail experiments clearly demonstrate that no buckling of mesoderm occurs when cell behaviors in the mesoderm are perturbed. Stating mesoderm buckles due to compression from the ectoderm is, therefore, misleading and has not been demonstrated with an experiment. Any mention of this interpretation should be confined to the discussion.

As elaborated in our response to Essential Revisions #6, we believe a model where apical constriction and ectodermal compression jointly mediate mesoderm buckling is compatible with the observation in the *snail* mutant embryos. In short, in our model, the level of compressive stress from the ectoderm is not sufficient to trigger mesoderm buckling on its own, but it can facilitate mesoderm buckling when the mesodermal cells undergo apical constriction. That being said, we agree with the reviewers that we have not tested this mechanism experimentally. In the revised manuscript, we emphasize that mesoderm buckling facilitated by ectodermal compression is a proposed mechanism that has only been tested by computer modeling.

8. In the discussion, the statement "Using computer modeling, we further demonstrated that mechanical bistablity in the mesoderm can arise from an apicobasal shrinkage of the ectoderm, which generates in-plane compression as the cell volume remains conserved" is misleading. This needs to be clarified, if the authors wish to raise the idea of bistability in the Discussion section.

Following this suggestion, we modified the sentence as following to clarify our point:

“Using computer modeling, we tested a possible mechanism based on the analogy of buckling of a compressed elastic beam induced by an indentation force. […] In this model, apicobasal shrinking of the ectoderm is expected to generate in-plane compression due to cell volume conservation.”

9. While the authors consistently claim precise agreement between the model and data, it remains unclear to what extend this is the case. Visuals of simulations are provided. But there are no quantitative comparisons found that directly compare a model result with a corresponding measurement. Therefore, such claims (e.g. page 14 "In particular, the transitional state of the tissue revealed in the simulation is nearly identical to that identified in our ontogenetic experiments") have to be toned down (e.g. looks visually similar).

Following the reviewers’ suggestion, we have modified the statement regarding the comparison between the modeling results and the actual observations in the embryos:

“In particular, the transitional state of the tissue revealed in the simulation is visually similar to that identified in our optogenetic experiments.”

10. The quantitative analysis shown in figure 8 appears inconsistent with the descriptions. First, the authors refer to a rate of volume reduction, but show volume. Moreover, rate of volume reduction in WT appears consistently different from snail, yet is described as very similar. Such strong claims should either be toned down or backed up with a statistical significance test.

We thank the reviewers for pointing this out. In the original text, we mentioned the “rate of volume reduction” one time in the following sentence: “In addition, the rate of Vol_ec_ reduction was similar between the wildtype and snail embryos, although the snail embryos on average had a smaller Vol_ec_ to start with (Figure 8d and f).”. In the actual plot, we displayed the volume over time, but did not display the rate of volume change directly. We apologize for the confusion.

To address this issue and also the reviewers’ comments regarding the comparison between the wildtype and snail mutants, we now provide quantification of the data to further support our statement. We fitted the descending part of the volume curve into straight lines and calculated the rate of volume reduction. The result is shown in Figure 8h in the revised manuscript. We found that there is no significant difference between the two groups (two-sample two tailed student t-test).

11. In the text page 20, the authors describe "In the wild type embryos, the compression promotes ventral furrow invagination, which in turn functions as a 'sink' to facilitate the movement of the ectoderm in the ventral direction". There is no experimental evidence provided for compression, and therefore this statement is speculation. Please rephrase.

Following the reviewers’ suggestion, we rephrased the description as follows:

“The altered pattern of cell movement in snail embryos could be explained by a combined effect of ectodermal compression and lack of ventral furrow invagination (Figure 8f). In this hypothetical scenario, the compression promotes ventral furrow invagination in wildtype embryos, which in turn functions as a “sink” to facilitate (and thereby bias) the movement of the ectoderm in the ventral direction.” (Page 22)

During the revision of this manuscript, we became aware of an earlier study that investigated the stage-specific role of Rok during ventral furrow formation by timed injection of Rok inhibitor (Krajcovic and Minden, 2012). In this work, the authors presented evidence that VF formation is less sensitive to injection of Rok inhibitor once the furrow started to form a small apical indentation. Due to technical limitations, the exact timing of Rok inhibition and its impact on the 3D morphology of the furrow were not provided in this study. However, their major observation is consistent with our results. We now cited this work in the Introduction of the revised manuscript (Page 4) to acknowledge the finding from this pioneer work.

Reviewer #1 (Recommendations for the authors):[…] (1) The manuscript offers no support for mechanical bistablity assumption. That purely elastic materials can buckle under compression is well established. But, the authors supply data that argues against mechanical bistablity in this system.– Experimental data does not go beyond correlation, and yet the mechanism presented claims a causal role of ectoderm compression for ventral furrow folding. The original manuscript attempted to provide experimental data in support of the mechanical instability model. The current figures describe cell shapes in 3D, but there is no test of causality offered. This is somewhat puzzling, as the optogenetic tool should also function in the lateral ectoderm.

As described in our response to Essential Revision #6, we agree with the reviewers that we could not entirely rule out other possible mechanisms that may result in the observed binary tissue response upon acute Rho1 inhibition. However, we consider mechanical bistability as the most parsimonious explanation of the observed binary tissue response. In addition, while it has been well established that elastic material can buckle under compression, the notion that tissue can undergo buckling through a joint action of global compression and local apical constriction and will display bistable characteristics during the folding process, to the best of our knowledge, have not been previously proposed. We therefore believe that such a model would provide new perspective to the understanding of tissue folding process.

To address the reviewer’s concern about the lack of causal role of ectodermal compression for ventral furrow folding, in the revised text, we clearly distinguish between the proposed model and the description of experimental observations.

– The model should be described as a theory. In its current form it is hard to distinguish from the descriptions of experiments. It should be clearly labeled that it is purely elastic, and that it neglects the well known viscous properties of tissues that dominate on the scale of at least 4 minutes and beyond.

Following reviewer’s suggestion, we carefully distinguished between descriptions of experimental observations and conclusions drawn from the modeling analysis. We have also emphasized the difference between the actual embryo mechanical properties and that of the model we used in our work, as elaborated in our response to Essential Revision #2.

– It is not clear to what extend the material in figure 8 supports the main argument. Instead, it seems to show the opposite. Lateral ectoderm shorting happens wether or not ventral furrow forms. This is a clear demonstration that the proposed mechanical bistability assumption is not able to drive tissue folding. Instead, these results suggest that folding needs to occur through an independent mechanism. Deepening of the fold could be generated by another mechanism (see below).

We thank the reviewer for raising this interesting and important point. As elaborated in our response to Essential Revision #7, we believe that compression alone is not sufficient to trigger tissue folding due to the high energy barrier between the two stable tissue configurations. The lack of tissue invagination in *snail* mutant is consistent with our notion that apical constriction is critical for letting the tissue to overcome this energy barrier and invaginate.

– Compression, as indicated by the authors, implies reduction in apical surface area of compressed cells, which is not shown in Snail or early opto RhoDN experiment. Early opto RhoDN experiments are described as heterogeneous cell morphology, but not further analyzed because of technical challenges. It is not clear how this is a technical problem, and not an issue of data interpretation. Heterogeneous apical cell area is consistent with cell shear, but not with external compression.

We agree with the reviewer that compression would be expected to result in a reduction in apical surface area of the compressed cells. The extent of this reduction would be jointly dependent on the magnitude of compression and the “resistance” – the rigidity of the cells. We did not directly measure apical surface area in *snail* mutant embryos due to lack of a membrane marker in the line we used. Based on our measurement of lateral cell movement, we anticipate that ventral cells will reduce their size mainly along the DV axis (since there is no sign of cell loss from the ventral surface). It would be interesting to test this in the future by direct measurements. For the early Opto-Rho1DN experiment, the apical cell area in the ventral region became heterogeneous ~6 min after the onset of gastrulation, which made it difficult to appreciate the cell aspect ratio change in individual cells. However, following the same reasoning for the *snail* mutant, we anticipate that the average cell aspect ratio (DV/AP) would reduce upon the observed ventral movement of the lateral cells. The cause of the heterogeneous apical cell area in the ventral region is unclear. Rho1 inhibition prevented activation of apical actomyosin contractility, which would be expected to cause disassembly of adherens junctions in ventral cells due to the expression of Snail (Weng and Wieschaus, 2016). This may in turn cause destabilization of the apical domain of the cells. We propose that what occurred in the ventral cells in early Opto-Rho1DN treated embryos was a combination of compression (as indicated by the ventral movement of the lateral cells, Figure 9f-i) and Snail-mediated loss of cell adhesion. It would be interesting to test these ideas in the future.

2) Discussion of model limitations and alternative scenarios.– I congratulate the authors on their observation in figure 4b. It seems reasonable to further analyze this interesting phenomenon, and study the possible impact of tissue tissue interactions. In doing so, the manuscript would benefit from an open approach.– The manuscript has valuable data on cell behaviors in the lateral ectoderm. But, the presentation is entirely focused around the idea of 'compression', and no alternatives discussed.– One such alternative, could be the impact of germband extension on deepening the already formed furrow. The cellular flow of germband has a component directed towards the ventral pole, possibly allowing cells to flow into furrow, that has already formed. This extension will lead to the observed apicobasal shortening of lateral ectoderm cells, and deepening of the fold, but requires no mechanical bistablity. It would further be consistent with lack of fold and the described cell shapes at the apical surface in Snail and opto RhoDN experiment.

We thank the reviewer for the suggestion and for raising the very interesting hypothesis about germband extension. As elaborated in our response to Essential Revision #6, our current data and previous published observations do not seem to support a major role of germband extension in triggering VF invagination. As suggested by the reviewer, we now included these discussions in the revised manuscript to broaden the discussion on the possible mechanisms that may lead to the binary tissue response observed in Figure 4.

– The timescale of elasticity is very short compared to the 20 minutes of ventral furrow. As pointed out by the authors, the cited paper by Doubrovinsky provides an estimate for the transition to viscosity within 4 minutes. It is one of the longest currently published timescales for this process. As the authors clearly demonstrated turnover of the actomyosin cytoskeleton is very fast, further indicating the four minutes estimate is an upper bound of what is to be expected for these cells.But, even if it is as long as four minutes, viscosity means, stresses will dissipate. Stress however is needed to build up for the proposed buckling by compression. Snail experiments clearly demonstrate that no buckling of mesoderm occurs when cell behaviors in the mesoderm are perturbed. Stating mesoderm buckles due to compression from the ectoderm is misleading, and has not been demonstrated with an experiment. Any mention of this interpretation should be confined to the discussion.

We thank the reviewer for raising the important question about the elasticity assumption. On the one hand, previous biophysical experiments suggest that the tissue display elastic response in the time scale of at least four minutes (Doubrovinski et al., 2017). On the other hand, rapid remodeling of cytoskeleton would be expected to rapidly dissipate elastic energy. Tissue buckling has been previously described in processes that have longer time scales than ventral furrow formation (Nelson, 2016). How observations at the molecular level link to the tissue level mechanical properties remains only partially understood and would be an important direction for future research. As elaborated in our response to the Essential Revision #7, we consider the proposed mechanism of tissue buckling by apical constriction and compression as the most parsimonious explanation for the binary tissue behavior we observed, but we also acknowledge that future experiments are needed to further support the proposed model. In the revised manuscript, we clearly indicate that buckling by compression is a proposed model that requires future experiments to further validate.

– In the discussion, the statement "Using computer modeling, we further demonstrated that mechanical bistablity in the mesoderm can arise from an apicobasal shrinkage of the ectoderm, which generates in-plane compression as the cell volume remains conserved" is misleading. The authors neither showed that ectoderm compresses mesoderm, nor is it a novel result that elastic systems can buckle under compression. This needs to be clarified, if the authors wish to raise the idea of bistability in the Discussion section.

Please see our response to the Essential Revision #7 for details.

– While the authors consistently claim precise agreement between the model and data, it remains unclear to what extend this is the case. Visuals of simulations are provided. But there are no quantitative comparisons found that directly compare a model result with a corresponding measurement. Therefore, such claims (e.g. page 14 "In particular, the transitional state of the tissue revealed in the simulation is nearly identical to that identified in our ontogenetic experiments") have to be toned down (e.g. looks visually similar).

Please see our response to Essential Revision #9 for details.

Reviewer #2 (Recommendations for the authors):Guo et al. have revised their paper by following the reviewer's suggestion.The science presented is now more solid and merits publications after addressing the following 4 points:1) In the new version of the paper, Guo and colleagues point out the fact that F-actin in ectoderm cells is not affected by RhoDN optogenetic activation. This is quite puzzling and therefore merits at least further discussion.

This is indeed a very interesting point. Please see our response to Essential Revision #4 for details.

2) In Figure 4A and in many other experiments, RhoDN is activated both in the mesoderm and in the ectoderm. Therefore, by following the logic of the authors model, ectoderm pushing is not dependent on Rho signaling. In other terms, while mesoderm cell shortening depends on Rho, ectoderm cell shortening is Rho independent. This also is quite surprising thus merits further discussion.

We thank the reviewer for this important comment. Please see our response to Essential Revision #5 for details.

3) For Figure 9f the authors have opted to track an ectoderm cell over time to demonstrate ectoderm displacement. This is fine, nevertheless the authors should be consistent and perform again the analysis for Figure 9f by using the same analysis protocol implemented for Figure 9a. More precisely the authors should follow a cell that is located 20 cells away from the midline (not just 10 cell away).

Following the reviewer’s suggestion, we have now tracked the cell movement in *snail* mutant embryos. Please see our response to Essential Revision #1 for details.

4) At page 19 and eventually in the discussion the authors should emphasize the discrepancy between the model and the in vivo measurements. More precisely they should make clear that along a cross-section in the model the minimum necessary ectoderm cell shortening is 5-10% for 60 cells while in the real embryo the cell-shortening measured is 4-8% for a much smaller number of ectoderm cells closer to the mesoderm (20 cells?).

We thank the reviewer for this suggestion. Please refer to our response to Essential Revision #3 for details.

References:

Denk-Lobnig M, Totz JF, Heer NC, Dunkel J, Martin AC. Combinatorial patterns of graded RhoA activation and uniform F-actin depletion promote tissue curvature. Dev Camb Engl 2021;148:dev199232. https://doi.org/10.1242/dev.199232.

Doubrovinski K, Swan M, Polyakov O, Wieschaus EF. Measurement of cortical elasticity in *Drosophila melanogaster* embryos using ferrofluids. Proc Natl Acad Sci U A 2017;114:1051–6.

Garcia De Las Bayonas A, Philippe J-M, Lellouch AC, Lecuit T. Distinct RhoGEFs Activate Apical and Junctional Contractility under Control of G Proteins during Epithelial Morphogenesis. Curr Biol CB 2019;29:3370-3385.e7. https://doi.org/10.1016/j.cub.2019.08.017.

Grevengoed EE, Fox DT, Gates J, Peifer M. Balancing different types of actin polymerization at distinct sites: roles for Abelson kinase and Enabled. J Cell Biol 2003;163:1267–79.

Homem CCF, Peifer M. Diaphanous regulates myosin and adherens junctions to control cell contractility and protrusive behavior during morphogenesis. Development 2008;135:1005–18. https://doi.org/10.1242/dev.016337.

Krajcovic MM, Minden JS. Assessing the critical period for Rho kinase activity during *Drosophila* ventral furrow formation. Dev Dyn Off Publ Am Assoc Anat 2012;241:1729–43. https://doi.org/10.1002/dvdy.23859.

Lye CM, Blanchard GB, Naylor HW, Muresan L, Huisken J, Adams RJ, et al. Mechanical Coupling between Endoderm Invagination and Axis Extension in *Drosophila*. PLoS Biol 2015;13:e1002292. https://doi.org/10.1371/journal.pbio.1002292.

Mason FM, Tworoger M, Martin AC. Apical domain polarization localizes actin-myosin activity to drive ratchet-like apical constriction. Nat Cell Biol 2013;15:926–36.

Nelson CM. On Buckling Morphogenesis. J Biomech Eng 2016;138:021005. https://doi.org/10.1115/1.4032128.

Polyakov O, He B, Swan M, Shaevitz JW, Kaschube M, Wieschaus E. Passive mechanical forces control cell-shape change during *Drosophila* ventral furrow formation. Biophys J 2014;107:998–1010.

Weng M, Wieschaus E. Myosin-dependent remodeling of adherens junctions protects junctions from Snail-dependent disassembly. J Cell Biol 2016;212:219–29. https://doi.org/10.1083/jcb.201508056.